# ECHOMOTION: UNIFIED HUMAN VIDEO AND MOTION GENERATION VIA DUAL-MODALITY DIFFUSION TRANSFORMER

**Yuxiao Yang**[1,2]  **Hualian Sheng**[2]  **Sijia Cai**[2,*]  **Jing Lin**[3]
**Jiahao Wang**[4]  **Bing Deng**[2]  **Junzhe Lu**[1]  **Haoqian Wang**[1,†]  **Jieping Ye**[2,†]
[1]Tsinghua University  [2]Alibaba Group  [3]Nanyang Technology University  [4]Xi'an Jiaotong University

## ABSTRACT

Video generation models have advanced significantly, yet they still struggle to synthesize complex human movements due to the high degrees of freedom in human articulation. This limitation stems from the intrinsic constraints of pixel-only training objectives, which inherently bias models toward appearance fidelity at the expense of learning underlying kinematic principles. To address this, we introduce EchoMotion, a framework designed to model the joint distribution of appearance and human motion, thereby improving the quality of complex human action video generation. EchoMotion extends the DiT (Diffusion Transformer) framework with a dual-branch architecture that jointly processes tokens concatenated from different modalities. Furthermore, we propose MVS-RoPE (Motion-Video Syncronized RoPE), which offers unified 3D positional encoding for both video and motion tokens. By providing a synchronized coordinate system for the dual-modal latent sequence, MVS-RoPE establishes an inductive bias that fosters temporal alignment between the two modalities. We also propose a Motion-Video Two-Stage Training Strategy. This strategy enables the model to perform both the joint generation of complex human action videos and their corresponding motion sequences, as well as versatile cross-modal conditional generation tasks. To facilitate the training of a model with these capabilities, we construct *HuMoVe*, a large-scale dataset of approximately 80,000 high-quality, human-centric video-motion pairs. Our findings reveal that explicitly representing human motion is complementary to appearance, significantly boosting the coherence and plausibility of human-centric video generation. Project page at: https://yuxiaoyang23.github.io/EchoMotion-webpage/.

## 1 INTRODUCTION

Recently, video generation models have witnessed a remarkable progress, driven in particular by the rapid evolution of diffusion models (Ho et al., 2020; Song et al., 2020; Peebles & Xie, 2023; Rombach et al., 2022) and VLM caption models (Liu et al., 2023a; Wang et al., 2024b; Team et al., 2024). Existing video generation models (Hong et al., 2022; Wan et al., 2025; Kong et al., 2024; Lin et al., 2024) have achieved promising results in terms of visual fidelity and temporal consistency. Nonetheless, even state-of-the-art generators still face significant challenges in synthesizing complex human motions. The resulting videos often suffer from severe anatomical artifacts and unnatural joint articulations, as exemplified in Figure 1(a). This deficiency primarily arises from the inherent limitations of pixel-only-regression training objectives, which tend to prioritize visual fidelity over the underlying kinematic principles governing human articulation. Concretely, the pixel-level reconstruction loss commonly used in diffusion models is dominated by static appearance and background details, not temporal kinetics. This deficiency is particularly pronounced for human subjects, whose high degrees of freedom make even subtle kinematic errors glaringly unnatural.

---

*Project leader.
†Corresponding author.

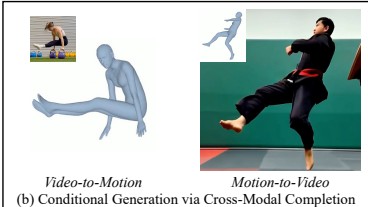

*Base model*  *EchoMotion*  *Video-to-Motion*  *Motion-to-Video*
(a) Enhanced Human Anatomical Generation  (b) Conditional Generation via Cross-Modal Completion

Figure 1: Overview of EchoMotion capabilities: (a) Improving anatomical integrity in human-centric video synthesis and (b) enabling bidirectional control between video and motion. By processing visual and motion sequences within a unified dual-branch Diffusion Transformer, the model learns a joint distribution of human appearance and kinematics.

Prior research (Chefer et al., 2025; Huang et al., 2024) has highlighted that human video synthesis problems persist even with basic motion types that are well-represented in training data. This finding echoes the insight that the lack of anatomical plausibility is not merely a matter of data scale, but rather points to inherent difficulties in modeling fine-grained kinematic dynamics. To address this issue, some existing works focus on conditioning generation on explicit structural guidance, employing techniques like 2D keypoints priors(Jiang et al., 2025; Ma et al., 2024b) or 3D poses priors(Zhou et al., 2024; Buchheim et al., 2025). While this approach offers direct control, it introduces two significant limitations. First, it creates a dependency on control signals that are often unavailable in real-world applications. Second, even when 3D priors like human poses are used, they are typically projected into the 2D image plane to align with the video frames. This projection process inevitably discards crucial 3D geometric information, leading to a diminished understanding of the underlying body structure and potential motion inconsistencies.

Motivated by these findings, we propose EchoMotion, an innovative framework that natively models the joint distribution of video and human motion modality. It features a dual-modality diffusion transformer, which adopts a dual-branch architecture to uniformly process tokens from different modalities. Unlike the previous MMDiT architecture (Esser et al., 2024), which solely focuses on denoising video inputs, EchoMotion takes a further step by explicitly denoising parametric motion jointly. Benefiting from this explicit motion modeling, our method can significantly reduce motion artifacts and facilitate the generation of complex human motion videos.

In contrast to common approaches that condition on motion videos (Zhou et al., 2024; Hu, 2024), our parameterized motion representation is not only more token-efficient but also preserves native 3D structural information, which is more conducive to the model learning kinematic patterns. Specifically, visual and motion tokens are concatenated along the sequence dimension and processed by our proposed Motion-Video Synchronized RoPE (MVS-RoPE) mechanism. This mechanism applies token-wise position embeddings to both modalities, enforcing precise temporal correspondence while preventing spatial positional collisions between the video and motion streams. To ensure efficient model convergence and fully exploit the multi-task capabilities of our model, we design a Motion-Video Two-Stage Training Strategy. This strategy employs a two-stage training recipe: an initial motion-only training phase followed by a motion-video multi-task training phase. In the second phase, we further construct three paradigms: joint generation, motion-to-video generation, and video-to-motion generation, to efficiently capture the cross-modal interactions. Furthermore, we introduce a new high-quality, human-centric video dataset, which we name *HuMoVe*. This dataset comprises approximately 80,000 video clips, each featuring distinct and clear human motions. For every video, we provide a granular textual description that details (1) the subject's appearance and attire, (2) the background context, and (3) a precise description of the action being performed. To enable multi-modal joint training, we also extract and include the corresponding SMPL motion parameters for each video clip, offering rich structural guidance.

Our experiments demonstrate that by holistically modeling the joint distribution of video and motion, EchoMotion significantly enhances the synthesis of human videos in terms of temporal coherence and structural integrity. Comprehensive evaluations validate its superiority over state-of-the-art baselines, showcasing drastic reductions in motion artifacts and superior preservation of physical plausibility. Furthermore, this unified approach inherently enables versatile cross-modal controllable generation, a capability confirmed through extensive qualitative and quantitative assessments.

## 2 RELATED WORKS

### 2.1 VIDEO DIFFUSION MODEL

Diffusion models have become the de facto standard for visual synthesis, scaling from high-fidelity image generation (Rombach et al., 2022; Esser et al., 2024; Peebles & Xie, 2023; Peng et al., 2024; Li et al., 2025; Wang et al., 2024a; 2025b) to video (Blattmann et al., 2023; Lin et al., 2024; Wan et al., 2025; Zhang et al., 2025; Wang et al., 2025c; Wu et al., 2025; Wang et al., 2025a) and 3D assets (Long et al., 2024; Xiang et al., 2025; Yang et al., 2025b; Liu et al., 2023b; Yang et al., 2025a; Zhao et al., 2025; Wang et al., 2025e;d). The core architecture, typically a Diffusion Transformer (DiT) (Peebles & Xie, 2023) operating in the latent space of a VAE, was originally designed for static images. By extending the 2D attention to a 3D full attention (Wan et al., 2025; Kong et al., 2024) or inserting additional temporal attention layers (Blattmann et al., 2023; Gao et al., 2025), the diffusion model is adapted for the temporal coherent video generation. Regarding conditioning, instead of using conventional cross-attention to inject text prompts, MMDiT (Esser et al., 2024) employs a distinct approach. It processes visual and textual tokens with separate weights and then concatenates them as input to the attention mechanism. This design facilitates a bidirectional flow of information between the two modalities. To further improve the temporal coherence, VideoJAM (Chefer et al., 2025) introduced to incorporate explicit motion prior to video models through predicting optical flow in addition to appearance. Instead of focusing on dense, low-level motion like optical flow to model short-term temporal motion, our work focuses on leveraging the SMPL (Loper et al., 2023) parameter as a high-level, structured human kinematics.

### 2.2 CONDITIONAL HUMAN VIDEO GENERATION

Recent advancements in diffusion models have significantly improved the quality of human video generation. Building upon pre-trained diffusion models, methods like DisCo (Wang et al., 2024c) and Follow-Your-Pose (Ma et al., 2024b) adapt the ControlNet architecture to guide generation using 2D human keypoints. Other works, such as MagicAnimate (Xu et al., 2024) and Animate Anyone (Hu, 2024), employ dedicated pose guidance modules to encode 2D pose sequences and inject them as conditioning. Another branch of research uses rendered 3D human models for conditioning. For instance, Champ (Zhu et al., 2024) utilizes rendered SMPL (Loper et al., 2023) models as guidance frames for video generation. Following this direction, RealisDance (Zhou et al., 2024; 2025) guides generation by concatenating multiple visual pose representations—such as outputs from HaMeR (Pavlakos et al., 2024) and DWPose (Yang et al., 2023), alongside rendered SMPL models—along the channel dimension. Similarly, Human4DiT (Shao et al., 2024) generates free-view human videos conditioned on rendered SMPL models and camera poses. While powerful, these methods share a fundamental architectural limitation: they are strictly conditional generators. In contrast, we introduce a unified architecture that treats human motion parameters and video as coupled modalities, enabling both joint generation and cross-modal completion.

## 3 METHOD

This paper introduces EchoMotion, a system designed to generate videos with corresponding motion sequences from an input text prompt. We first propose a **Dual-Modality Diffusion Transformer**, augmented with a parametric motion representation and a **Motion-Video Synchronized RoPE**, to effectively model the intricate interaction between these two distinct modalities in Sec. 3.1. Then, we tailor a **Multi-Modal Two-Stage Training Strategy** to facilitate mutual promotion and completion within this multi-modal system, as detailed in Section 3.2. Finally, to support this research and the broader community, we construct and release the *HuMoVe* dataset, a large-scale collection of paired video, 3D human motion parameters, and text data in Section 3.3.

### 3.1 DIFFUSION FOR JOINT VISUAL-MOTION GENERATION.

**Architecture.** Towards high-fidelity video generation, we select Wan (Wan et al., 2025) as our backbone model, owing to its superior performance. Unlike approaches that model the video-only distribution $p(x|y)$-which can prioritize appearance fidelity at the expense of motion principles, we instead model the joint distribution of human movement and video $p(x, m|y)$. Here, $y$ denotes the

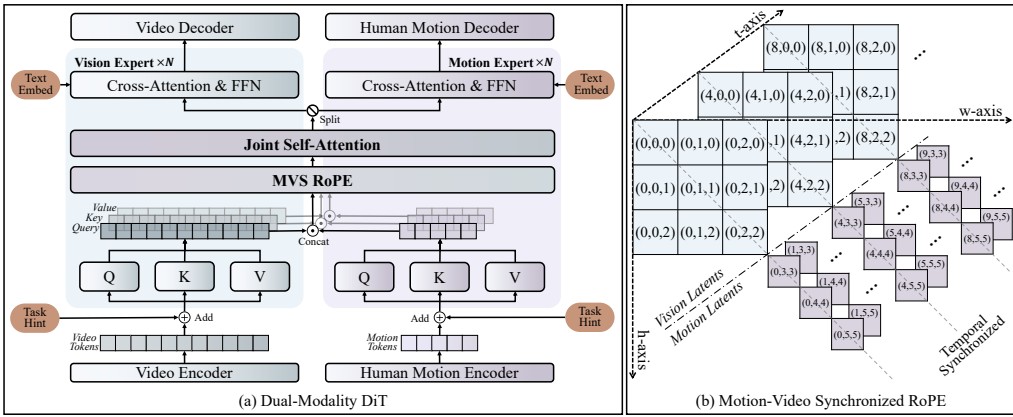

Figure 2: Overview of EchoMotion. (a) The dual-modality DiT block for joint video-motion modeling. (b) Our MVS-RoPE to serve as a synchronized coordinate for dual-modal token sequence.

given text prompt, while $x$ and $m$ represent the video and human motion parameters, respectively. This unified modeling approach allows our model to effectively learn both visual and motion dynamics. As illustrated in Figure 2, our Dual-Modality Diffusion Transformer first processes the input video by parameterizing human motion representations, derived from SMPL pose and shape parameters (Loper et al., 2023). These motion tokens are concatenated with visual tokens to form a **unified multi-modal context sequence**, which is then fed into a series of Dual-Modality Diffusion Transformer blocks. Within each block, our proposed MVS-RoPE encodes the precise position of each token within the unified multi-modal context. This ensures that during the joint self-attention process, both intra-modal and cross-modal information is exchanged properly.

**Parametric Human Motion Representation.** Given a video of a person in motion, we use the SMPL (Loper et al., 2023) model to parameterize the pose and shape of the human body. It captures articulated human body configurations through a low-dimensional set of pose and shape parameters, making it a widely adopted foundation in computer vision tasks involving human mesh recovery. For a single frame, the SMPL model facilitates the extraction of human body representations. Specifically, it provides shape parameters $\beta \in \mathbb{R}^{10}$ that define the overall body shape, pose parameters $\theta \in \mathbb{R}^{24 \times 6}$ that capture human joint angles, global body orientation $\gamma \in \mathbb{R}^6$, and the human root joint position $v \in \mathbb{R}^3$. Following DART (Zhao et al., 2024), we further utilize the 3D joint position $\eta \in \mathbb{R}^{24 \times 3}$ to represent each human joint.

To construct a unified representation that integrates human motion modalities, we design multi-head projectors to build a bidirectional mapping between the parameter and the latent spaces. For each frame, we categorize the motion parameters into three groups: $\{v, \eta\}$ for 3D position, $\{\theta, \gamma\}$ for 6D rotation, and $\beta$ for human shape. Three independent MLPs then project these parameters sets into the target transformer hidden dimension, generating 51 motion tokens per frame. Similarly, another three MLPs map the generated motion tokens back to the original parameter space for reconstruction. Crucially, to better model rapidly changing motion patterns, we preserve the temporal structure of these motion tokens. This contrasts with the typical time-level down-sampling applied to visual tokens, and importantly, these refined and compact motion tokens retain crucial temporal motion information with only a minimal increase in computational overhead.

**Dual-Modality Diffusion Transformer Block.** This block processes and integrates multi-modal information. It begins by passing the embeddings from the video and human motion modalities through modality-specific projections, implemented as two distinct sets of learnable matrices. The projected features are then concatenated along the sequence dimension to form as:

$$Q_{mm}, K_{mm}, V_{mm} = [Q_v; Q_m], [K_v; K_m], [V_v; V_m], \tag{1}$$

where $[\cdot ; \cdot]$ signifies the sequence-level concatenation operation. Thereafter, a joint self-attention layer is applied to capture dependencies and correlations across both modalities. Following self-attention, the attended features are disentangled, with each modality's features subsequently processed independently through separate cross-attention layers (to interact with text information) and FFNs. This architecture enables a detailed interaction between modalities and with textual guidance.

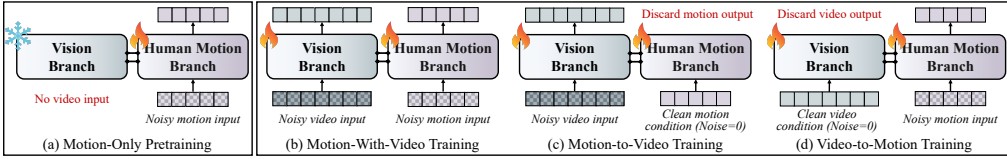

Figure 3: Overview of our Motion-Video Two-Stage Training Strategy. In Phase 1, the model is pretrained on motion-only data. In Phase 2, we conduct multi-task training on paired motion-video data, regarding "motion-with-video", "motion-to-video", and "video-to-motion" as three distinct tasks to be learned simultaneously.

Within self-attention layers, we propose a specialized multi-modal positional embedding to inject features with precise position information. This enables the model to effectively reason about token relationships both within and across modalities, as formulated below.

**Motion-Video Synchronized RoPE.** Existing MMDiT (Esser et al., 2024; Kong et al., 2024) architectures typically incorporate positional information either through inherent positional IDs from the text encoder or by employing M-RoPE (Wang et al., 2024b), which extends relative positional encoding to jointly handle both text and visual modalities in a unified sequence. However, neither of these approaches directly accounts for the inherent temporal alignment between human motion and videos. Consequently, they are not directly suitable for representing the positional information of parameterized motion tokens, necessitating a dedicated design to accurately capture this crucial motion positional information.

As illustrated in Figure 2(b), our MVS-RoPE is designed to handle the distinct spatio-temporal nature of our multi-modal context sequence, which is composed of both video and motion latent tokens. Spatially, we partition the coordinate space: video tokens occupy a base $(h, w)$ region, while motion tokens are treated as a "diagonal extension" by offsetting their spatial index. This ensures modality distinction. Temporally, recognizing the direct temporal correspondence between motion tokens and visual tokens (4x temporal compression from video VAE), we assign a scaled indexing scheme. While video tokens are indexed by time $t$, the corresponding coarse-grained motion tokens are assigned a scaled index of $t/4$. This directly embeds their temporal synchrony into the positional encoding. The processed feature after MVS-RoPE is given by:

$$\hat{\boldsymbol{f}}_{t,h,w}^v = \text{MVS-RoPE}(\boldsymbol{f}_{t,h,w}^v, t, h, w) = \mathcal{R}(t, h, w) \cdot \boldsymbol{f}_{t,h,w}^v, \tag{2}$$

$$\hat{\boldsymbol{f}}_{t,i}^m = \text{MVS-RoPE}(\boldsymbol{f}_{t,i}^m, t, i) = \mathcal{R}(\frac{1}{4}t, H + i, W + i) \cdot \boldsymbol{f}_{t,i}^m. \tag{3}$$

Here, $t$ is the temporal index, $i$ is the motion token index, and $(h, w)$ represents the spatial indices for visual tokens. $H$ and $W$ denote the spatial range of the visual tokens. The function $\mathcal{R}(\cdot)$ represents the RoPE encoding, which applies a rotation based on the input temporal and spatial indices.

This design ensures that 1) Preserves pretrained knowledge: video tokens receive the exact same 3D RoPE as used during pre-training, ensuring that valuable learned representations are not disturbed; 2) Enforces correct temporal alignment: the 1/4 scaling explicitly encodes the multi-rate relationship, resolving ambiguity and ensuring video and motion are perfectly synchronized in time. and 3) Guarantees modality distinguishability: the spatial diagonal extension prevents "positional collisions", allowing the model to easily differentiate between video and human motion tokens.

## 3.2 MOTION-VIDEO TWO-STAGE TRAINING STRATEGY.

Due to the divergent feature representations of visual and motion tokens, a phased training approach is necessary for the model to effectively process and align these two modalities. To achieve this alignment and ensure stable learning, we propose a two-stage strategy initialized from pretrained DiT parameters:

- **Phase 1: Motion-only Pretraining.** The motion branch is trained independently using motion-only datasets, while the video branch is frozen and deactivated (inputs omitted). This stage focuses on generating motion sequences.

Figure 4: Overview of our *HuMoVe* dataset. (a) Voronoi treemap of the dataset's composition. (b) Word cloud of the text captions. (c) Sample frames paired with their 3D mesh reconstructions.

- **Phase 2: Motion-video Multi-task Training.** Subsequently, the model is trained on motion-video paired datasets with both branches unfrozen and active, enabling the generation of both visual and motion sequences.

The rationale for motion-only pretraining is to allow the motion branch to first converge on its own domain, preventing the dominant computationally expensive of video branch. Once stable, the architecture is readily extensible to both joint generation of video and motion, as well as bi-directional cross-modal control. Therefore, in phase 2, we train the model using motion-video paired datasets with a focus on the three complementary interaction paradigms.

As illustrated in Figure 3, each paradigm is randomly sampled: 1) Joint training: generate both video and motion sequences concurrently. 2) Motion-to-video training: motion sequences serve as the conditioning input for video generation. 3) Video-to-motion training: video sequences are used to condition motion generation. When one modality serves as the conditioning input, its features are preserved and not subjected to noise injection during the forward diffusion process. Additionally, a lightweight MLP projects the task embedding to the latent space. This task hint is then added to the latents to guide conditional token prediction. Under this framework paradigm, the model naturally achieves both pure text-guided motion-video generation and cross-modal conditional generation during inference.

**In-Context Classifier-Free Guidance (ICCFG).** Our ICCFG implementation employs distinct conditioning strategies tailored to our three training paradigms in Phase 2, differing from standard CFG. During Phase 2, we apply a paradigm-specific conditional dropping strategy. For joint generation, the text condition is randomly dropped. For motion-to-video generation, both text and motion conditions are randomly dropped. For video-to-motion generation, the text condition is always dropped, while the video condition is dropped randomly. Accordingly, to leverage this capability during inference, we define $\boldsymbol{u}_\theta(\cdot)$ as the diffusion model. For the Joint Generation mode:

$$\boldsymbol{o}_t^v, \boldsymbol{o}_t^m = \boldsymbol{u}_\theta(x_t, m_t, \emptyset) + \omega_1(\boldsymbol{u}_\theta(x_t, m_t, y) - \boldsymbol{u}_\theta(x_t, m_t, \emptyset)), \tag{4}$$

where $\boldsymbol{o}_t^v$ and $\boldsymbol{o}_t^m$ represent the video and motion predictions of timestep $t$, respectively. $\emptyset$ signifies the absence of a specific condition and $\omega_1$ is the guidance scale for text condition. For the Motion-to-Video Generation mode, the output can be expressed as:

$$\boldsymbol{o}_v^t = \boldsymbol{u}_\theta(x_t, \emptyset, \emptyset) + \omega_1(\boldsymbol{u}_\theta(x_t, m_t, y) - \boldsymbol{u}_\theta(x_t, m_t, \emptyset)) + \omega_2(\boldsymbol{u}_\theta(x_t, m_t, \emptyset) - \boldsymbol{u}_\theta(x_t, \emptyset, \emptyset)), \tag{5}$$

where $\omega_2$ is the guidance scale for motion condition. For the Video-to-Motion Generation mode:

$$\boldsymbol{o}_t^m = \boldsymbol{u}_\theta(\emptyset, m_t, \emptyset) + \omega_2(\boldsymbol{u}_\theta(x_t, m_t, \emptyset) - \boldsymbol{u}_\theta(\emptyset, m_t, \emptyset)). \tag{6}$$

This allows the model to leverage the rich context provided by various modalities while selectively enabling specific generation capabilities.

### 3.3 *HuMoVe* DATASET

A significant gap exists in current open-source resources for joint video-motion generation. Datasets from prior work (Mahmood et al., 2019; Lin et al., 2023; Fan et al., 2025) are often unsuitable, typically because they are modality-specific (lacking visual context) or consist of low-quality videos with redundant backgrounds and human characters. To address this limitation, we construct *HuMoVe*: a large-scale, high-quality dataset featuring diverse scenes and human characters. Each video

Table 1: Comparison with baseline models on video generation. We report both human evaluation and automatic metrics. **Bold** indicates the best performance, and underline indicates the second best.

| | Auto Metrics | | | | Human Eval | | |
|---|---|---|---|---|---|---|---|
| | Human Anatomy | Motion Smoothness | Dynamic Degree | Aesthetic Quality | Video Quality | Prompt Following | Posture Plausibility |
| CogVideoX-2B | 61.7 | 97.0 | 49.4 | 51.6 | 55.3 | 52.1 | 53.6 |
| Wan-1.3B | 78.1 | 98.2 | 60.6 | **60.1** | 68.2 | 70.3 | 64.0 |
| Video Tuning(Wan-1.3B) | 77.4 | 98.3 | 61.6 | 59.7 | 69.3 | 73.2 | 65.5 |
| EchoMotion(Wan-1.3B) | 79.6 | 98.9 | 61.9 | 60.0 | 71.3 | 73.2 | 66.1 |
| CogVideoX1.5-5B | 65.3 | 98.5 | 54.4 | 53.2 | 62.5 | 60.4 | 59.4 |
| Wan-5B | 83.0 | 98.9 | 62.2 | 58.3 | 72.8 | 78.9 | 68.9 |
| Video Tuning(Wan-5B) | 83.1 | 98.7 | 63.1 | 57.9 | 72.3 | 79.6 | 70.2 |
| EchoMotion(Wan-5B) | **85.1** | **99.3** | **64.0** | 58.3 | **81.0** | **81.5** | **81.6** |

*"The video shows a skateboarder quickly sliding down a stair railing, maintaining balance throughout, with bold and smooth movements."*

*"A gymnast is stretching on the mat before training, lifting her legs over her head and showing off her incredible flexibility."*

*"A fit, muscular female athlete is performing a dynamic workout combining squats and cross touches in an indoor space. She is in a standing position, with her hands clasped in front of her chest. Then, she quickly squats to complete one squat movement. Immediately after, she lifts one leg high to complete a stretching movement…"*

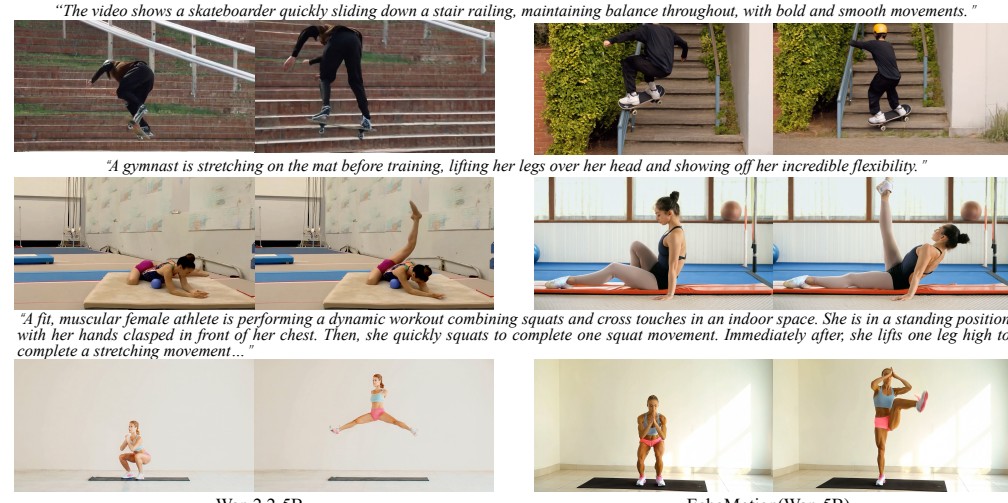

Wan 2.2-5B             EchoMotion(Wan-5B)

Figure 5: Qualitative comparison with the 5B baseline. Our model (EchoMotion, right) generates anatomically correct and semantically coherent human motions, resolving the severe artifacts and compositional failures present in the baseline (left).

in *HuMoVe* is paired with a descriptive textual caption and its corresponding 3D SMPL motion parameters, forming a tightly aligned, multi-modal corpus for advanced generative modeling.

We implement a comprehensive data processing pipeline that involves automated collection from various sources, followed by rigorous filtering for aesthetic quality and subject focus, and culminates in precise 3D human mesh recovery and annotation, resulting in a final dataset of approximately 80,000 entries. A detailed breakdown of our pipeline is provided in Appendix A.4. The resulting *HuMoVe* dataset is exceptionally diverse. As visualized by the Voronoi treemap in Figure 4(a), it spans 9 major categories (represented by distinct color groups) and 38 subcategories (represented by shades within each group), ranging from subtle, everyday actions to dynamic, complex performances. A full categorical breakdown is available in the Appendix A.4. This careful curation provides a clean, comprehensive, and challenging foundation for developing kinematically-aware generative models.

## 4 EXPERIMENTS

### 4.1 IMPLEMENTATION DETAILS

We perform experiments on two variants of the open-sourced base model, Wan2.1-1.3B and Wan2.2-5B, to validate the effectiveness of our method. The initial motion-only training phase leverages a composite motion dataset of our proposed *HuMoVe* dataset alongside the extensive HumanML3D dataset (Guo et al., 2022), a public repository containing over 14,616 human motions in SMPL

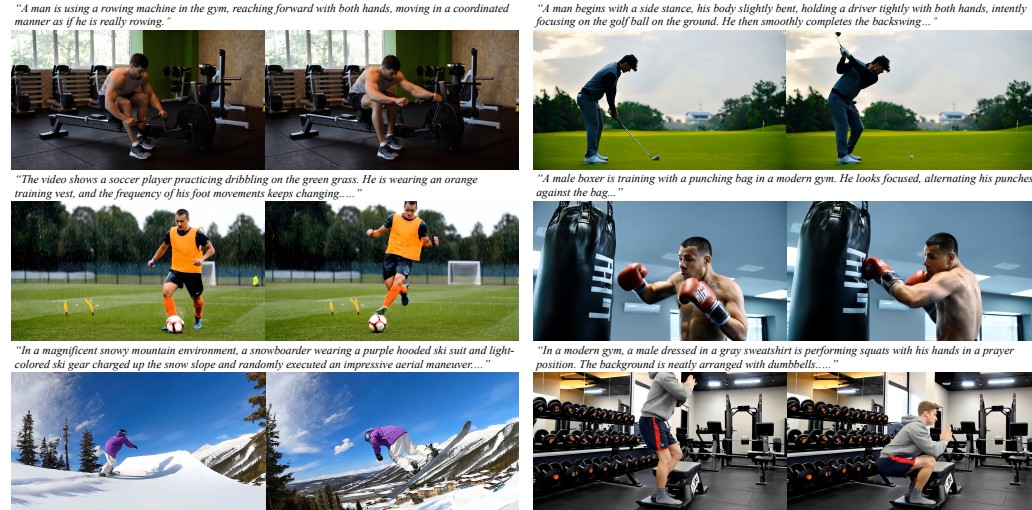

Figure 6: Text-to-video results from EchoMotion, demonstrating both strong prompt alignment and high kinematic plausibility across a diverse range of human-centric scenarios. Please refer to Appendix A.6 and the supplementary material for additional examples and video results.

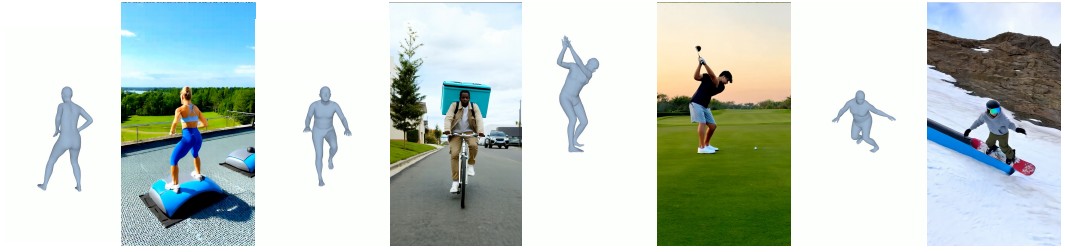

Figure 7: EchoMotion jointly generates an SMPL motion sequence (left) and video (right), demonstrating a learned joint distribution.

format, and is conducted for 15k steps. Following this, the motion-video multi-task training phase is performed on our collected motion-video paired data for an additional 12k steps. These step counts were determined empirically by monitoring both loss convergence and qualitative quality on a held-out validation set. The Wan2.2-5B+EchoMotion model is trained for 4000 A100 GPU hours. The whole training procedure is conducted on 32 NVIDIA A100 GPUs for about 4 days.

## 4.2 TEXT TO VIDEO GENERATION

**Evaluation Protocol.** To enable a comprehensive evaluation, we build a new benchmark that covers a wide spectrum of human motion, from daily activities to extreme athletic feats. Our benchmark includes a diverse set of prompts (30 per category) covering: precise, high-momentum movements from gymnastics and athletics; fluid, expressive motions from dance; reactive and interactive scenarios from ball and combat sports; and the natural gestures of everyday life. For quantitative analysis, we use the automatic metrics from VBench (Huang et al., 2024) and VBench-2.0 (Zheng et al., 2025), alongside human user studies to collect numerical ratings. These quantitative findings are complemented by qualitative visualizations that illustrate the performance of our method.

**Quantitative Results.** We evaluate our method against the video-only baselines at both 1.3B and 5B scales with baseline open-source models. The results, summarized in Table 1, demonstrate that our joint modeling approach yields substantial gains in motion fidelity. On automatic metrics, EchoMotion markedly improves scores for Motion Smoothness and Anatomical Consistency over the baselines. Notably, this improvement is achieved without any reduction in the Aesthetic Quality score. These results are strongly corroborated by our human evaluation, where participants

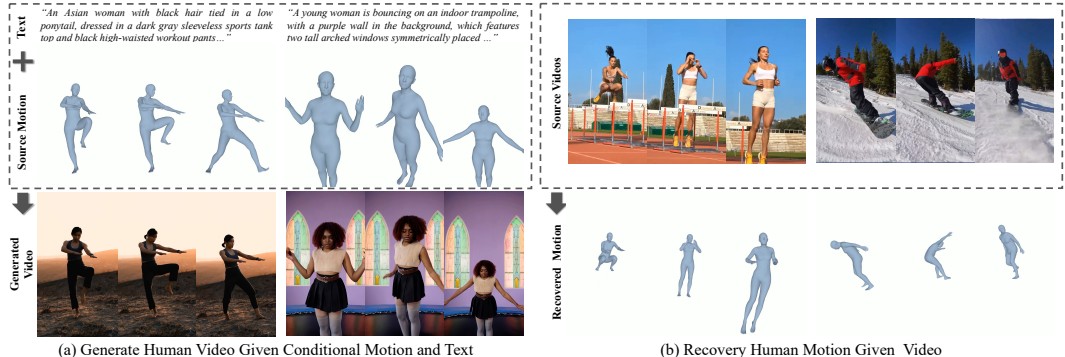

(a) Generate Human Video Given Conditional Motion and Text      (b) Recovery Human Motion Given Video

Figure 9: Cross-modal completion by EchoMotion. **(a)** Motion-to-Video synthesis from motion and text. **(b)** Video-to-Motion recovery (inverse kinematics).

assign EchoMotion significantly higher ratings for Video Quality, Motion Plausibility, and Prompt-Following for human-centric prompts.

**Qualitative Results.** Figure 5 highlights the critical limitations of Video-only training on complex, human-centric prompts. While the Wan2.2-5B baseline generates reasonable aesthetics, it consistently violates kinematic constraints, producing severe anatomical artifacts (e.g., the tangled gymnast, the distorted skateboarder). Furthermore, the baseline struggles with semantic compositionality, failing to execute the multi-step workout sequence. In contrast, by jointly modeling video and human motion, our model generates subjects with plausible anatomy and successfully follows compositional instructions. Our model generates coherent and physically plausible motions across all challenging cases, demonstrating that joint distribution modeling leads to a superior internal representation of human kinetics.

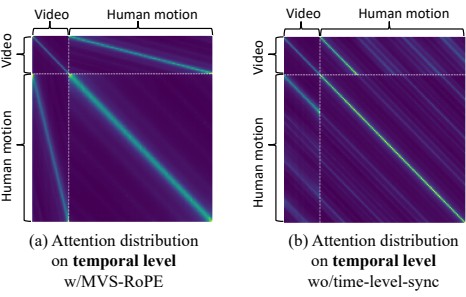

(a) Attention distribution on **temporal level** w/MVS-RoPE

(b) Attention distribution on **temporal level** wo/time-level-sync

Figure 8: Effect of MVS-RoPE.

In addition to outperforming baselines, our method demonstrates remarkable versatility and a sophisticated capacity for multi-modal generation. Figure 6 showcases its generative breadth, successfully synthesizing a wide variety of high-quality, kinematically plausible actions—from a golf swing to a snowboarding aerial—across diverse scenes. Moreover, the efficacy of our joint modeling approach is directly illustrated in Figure 7, where the model simultaneously generates a video and its corresponding, temporally-aligned SMPL sequence from one prompt. This co-generation demonstrates that the model's output is not simply a pixel-level synthesis but is fundamentally conditioned on an internal representation of human kinematics.

## 4.3 CROSS-MODAL COMPLETION

Our model's unified design enables powerful cross-modal capabilities, as shown in Figure 9. By forming tasks as modal completion, EchoMotion operates bi-directionally: (a) it can synthesize a high-fidelity video that precisely follows a given motion sequence (motion-to-video), and (b) it can recover the underlying SMPL motion from an input video (video-to-motion). This flexibility to perform both generation and inverse kinematics with a single model highlights the significant advantage of our joint modeling approach. We provide further quantitative evaluations for cross-modal completion tasks in Appendix A.7.2 and A.7.3.

## 4.4 ABLATION STUDIES

In this section, we conduct ablation studies to validate the key components of our framework. Additional ablations are detailed in Appendix A.8.

**Joint Modeling vs. Video-Only Modeling.** We compare our joint training approach against a baseline fine-tuned exclusively on the video data from our dataset (Video Tuning). As shown in Table 1, while the Video-only tuning approach offers only marginal improvements and fails to enhance key kinematic metrics, our EchoMotion model demonstrates substantial gains in both Human Anatomy and, notably, Posture Plausibility. This result confirms our central hypothesis: The key to high-quality human motion synthesis lies in the joint modeling of appearance and kinematics during training, whereas simply adding more human-centric video data offers marginal benefits.

**MVS-RoPE Design.** To validate our MVS-RoPE design for aligning video and motion modalities, we visualize the self-attention score in Figure 8. Crucially, the motion temporal sequence is four times the length of the video temporal sequence, requiring the model to learn a non-trivial 4:1 temporal mapping. In our model with MVS-RoPE (a), the attention map reflects this perfectly. The video-to-motion attention forms a clear, shallow diagonal, while the Motion-to-Video attention forms a corresponding steep diagonal. This asymmetrical structure is direct evidence that the model has learned the correct temporal alignment. In contrast, the baseline without MVS-RoPE (b) fails completely; the attention is scattered, and the required diagonal structures are absent, demonstrating its inability to synchronize the two modalities.

## 5 CONCLUSION

We present EchoMotion, an innovative framework for generating human videos by jointly modeling appearance and kinematics within a Dual-Modality DiT. Our approach utilizes a SMPL-based parametric representation for human motion, concatenating its tokens with video tokens into a unified multi-modal sequence. To enable effective information exchange within this sequence, we introduce MVS-RoPE, a novel positional embedding that enforces temporal alignment between video and motion modalities during multi-model joint self-attention. Furthermore, a Motion-Video Two-Stage Training Strategy endows the model with versatile, bi-directional capabilities: controllable motion-to-video and video-to-motion generation. The development of this framework is supported by our introduction of *HuMoVe*, a new large-scale dataset comprising approximately 80,000 high-quality video-motion pairs.

**Limitations.** Our framework is currently limited to single-person generation. Extending it to multi-person scenarios, while architecturally feasible by concatenating SMPL tokens, would require creating a new, large-scale dataset with per-person annotations. As this represents a significant resource commitment, we prioritized perfecting the single-person generation model in this work and leave multi-person generation as a promising direction for future projects.

ACKNOWLEDGMENT

This work is supported by the NSFC fund (62576190), in part by the Shenzhen Science and Technology Project under Grant (KJZD20240903103210014, JCYJ20220818101001004)

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

## A APPENDIX

### A.1 USE OF LLMS

We utilized Large Language Models (LLMs) in several capacities to assist in this research. The core intellectual contributions, including the formulation of the research problem, the design of the EchoMotion architecture, the MVS-RoPE mechanism, and the mixed multi-modal in-context learning strategy, were conceived and developed entirely by the human authors. The roles of LLMs were confined to the following assistive tasks:

1. **Writing and Manuscript Refinement:** We employed general-purpose LLMs, such as OpenAI's ChatGPT, for grammatical corrections, stylistic improvements, and enhancing the clarity and readability of the manuscript. The scientific narrative, structure, and all claims remain the original work of the authors.

2. **Data Curation Support:** To aid in the collection of our *HuMoVe* dataset, we used an LLM to generate a broad set of keywords and search queries related to complex human motions. This helped to systematically identify relevant video content from public-domain sources.

3. **Automated Video Annotation:** A significant part of the annotation process for the *HuMoVe* dataset was facilitated by Qwen-VL-Narrator (qwe). This model was tasked with generating the initial draft of the granular textual descriptions for each video clip, covering (1) the subject's appearance and attire, (2) the background context, and (3) a description of the action.

### A.2 PRELIMINARY

**Diffusion Tranformers(DiT).** Diffusion Transformers (DiTs) (Peebles & Xie, 2023) replace the conventional U-Net backbone in diffusion models with a pure transformer architecture, demonstrating superior performance and scalability. A DiT operates on latent space and is composed of three main stages:

- An input encoder that "patchifies" the noisy latent variable into a sequence of tokens.
- A series of transformer blocks that process these tokens. These blocks are conditioned on the diffusion timestep $t$ and other context, such as text embeddings, via adaptive layer normalization (adaLN).
- A final decoder that "unpatchifies" the output tokens back into the predicted noise map.

**Flow Matching.** During training, video latents $x_i$ is disturbed by a random noise $x_0 \sim \mathcal{N}(0, I)$, according to the timestep $t \in [0, 1]$, *i.e.*,

$$x_t = tx_1 + (1-t)x_0. \tag{7}$$

The model is trained to predict the velocity $v_t = x_1 - x_0$ by minimizing the MSE loss of the model prediction and $v_t$,

$$\mathcal{L} = \mathbb{E}_{x_0, x_1, y, t} ||u(x_t, y, t; \theta) - v_t||^2, \tag{8}$$

where $y$ is the input text description, $\theta$ denotes the model weights, and $u(x_t, y, t; \theta)$ is the prediction by the DiT model.

Table 2: Mapping of Major Categories to their corresponding Fine-grained Categories.

| Major Category | Fine-grained Categories |
|---|---|
| Urban & Outdoor | Archery, Cycling, Equestrian, Highlining, Motocross, Parkour, Rock Climbing, Skateboarding, Street Workout |
| Water Sports | Kayaking, Kitesurfing, Paddleboarding, Surfing, Swimming, Wakeboarding, Windsurfing |
| Athletics & Fitness | Athletics, Olympics |
| Ball Sports | Basketball, Golf, Soccer, Tennis, Volleyball |
| Combat Sports | Boxing, Judo, MMA, Taekwondo, Wrestling, Wushu |
| Dance & Artistic | Artistic Roller Skating, Breakdancing, Dancing |
| Gymnastics & Acrobatic | Gymnastics, Trampoline |
| Ice & Snow Sports | Figure Skating, Skiing & Snowboarding |
| General & Others | Daily Activity, Other |

**Rotary Position Embedding (RoPE).** Rotary Position Embedding(Heo et al., 2024) encodes absolute positional information by applying position-dependent rotations to the query and key vectors, which are typically adopted by recent Transformers to inject positional information within self-attention layers. Given a $d$-dimention vision embedding $x_{t,h,w}$ with a 3D positional index $(t, h, w)$ in the sequence, 3D RoPE is added dimension-wise,

$$\tilde{x}_{t,h,w} = R_{t,h,w}^{\Theta} x_{t,h,w},$$

(9)

where $R_{t,h,w}^{\Theta}$ denotes a 3D rotation matrix determinded by the given 3D positional index and a pre-defined base frequency $\Theta$.

## A.3 The Joint Distribution of Video and Human Motion.

Instead of modeling the video-only distribution, we propose that the distribution of human movement video, denoted as $p_\theta(z)$, can be modeled by a joint distribution of its video and corresponding human motion parameters. Specifically, given a text prompt $y$, the distribution $p(z|y)$ can be formulated as

$$p(z|y) = p_\theta(x, m|y),$$

(10)

where $x$ and $m$ represent the video and the human motion parameters, respectively. The goal of our model $f$ is to learn the joint distribution $p_\theta(x, m|y)$. During inference, we generate a video and corresponding human motion parameters by iteratively denoising the video and human motion latents jointly.

$$x, m = f(y)$$

(11)

Revisiting Eq.10, we could derive that

$$p_\theta(x|m, y) = \frac{p_\theta(x|y)}{p_\theta(x, m|y)},$$

(12)

where $p_\theta(x|m, y)$ corresponds to the video conditional distribution given the human motion. Therefore, sampling the video from the $p_\theta(x|m, y)$ is equivalent to generating a video that matches the given human motion $m$ and text prompt $y$, *i.e.*, controllable video generation. Similarly, given a video, we can recover the human motion parameters within the video by sampling from the motion conditional distribution:

$$p_\theta(m|x, y) = \frac{p_\theta(m|y = \emptyset)}{p_\theta(x, m|y = \emptyset)}.$$

(13)

## A.4 Details of *HuMoVe* Dataset

**Fine-grained Categories.** Table 2 details the two-level classification scheme used to organize our dataset. We group specific, fine-grained activities (e.g., Skateboarding, Swimming, Boxing) into

broader, thematic major categories (e.g., Urban & Outdoor, Water Sports, Combat Sports). This taxonomy provides a structured framework for analyzing the dataset's diversity and for evaluating model performance across different motion types.

**Raw Data Collection.** To ensure our raw data encompasses a wide range of human movements, our collection process began with a structured, top-down approach. We first manually defined nine major motion categories, as detailed in Table 2 (e.g., Urban & Outdoor, Water Sports, and Combat Sports). These high-level categories were then provided as input to a Large Language Model (LLM) (gim), which we tasked with generating a diverse list of specific search keywords for each category. An example of the prompt used for this task is shown below. The resulting keywords were subsequently used to collect a raw data pool from diverse sources, including open-source datasets, movies, and the internet. The prompt used for this task is shown in Listing 1.

```
You are a data sourcing expert for computer vision research. Your task is
    to expand high-level categories of human motion into specific,
    diverse, and fine-grained search keywords suitable for video
    platforms.

For each major category provided, generate a list of 20-30 search terms.

Here is an example of the desired input-output format:

**Input:**
Major Category: Combat Sports

**Desired Output Keywords:**
- boxing training drills 4k
- MMA sparring session slow motion
- cinematic Taekwondo high kick
- Judo throw tutorial
- female wrestler practice highlights
- Wushu performance competition
...

Now, please generate keywords for the following list of major categories:
- Urban & Outdoor
- Water Sports
- Athletics & Fitness
- Ball Sports
- Combat Sports
- Dance & Artistic
- Gymnastics & Acrobatics
- Ice & Snow Sports
- General & Others
```

Listing 1: The prompt used to expand major categories into specific search keywords via an LLM.

**Data Filtering.** Raw video data often exhibits frequent scene translation and contains low aesthetic and resolution data, which will negatively impact the performance of the model. To mitigate these challenges, we employ a scene detection model to segment the raw video into clips and leverage an aesthetic scoring model to filter out video clips with low aesthetic quality. As this work primarily focuses on the movement of the main characters in the foreground, we employed a bounding box detection model to identify the individuals in the video, and filtered out those containing multiple main characters by comparing the number and area of the bounding boxes. Moreover, the DWPose(Yang et al., 2023) is employed to detect the number of visible key points within the video. Those data with a relatively low number of visible key points are filtered to avoid artifacts and jitter during the 3D human pose detection stage.

**Human Mesh Recovery and Post Process.** Given the track bounding boxes as input, to obtain frame-wise high-quality SMPL parameters including $\beta \in \mathbb{R}^{10}$, $\theta \in \mathbb{R}^{24}$, $\Gamma \in \mathbb{R}^{6}$, and $v \in \mathbb{R}^{3}$, we use the CameraHMR(Patel & Black, 2025) for the human mesh estimation, due to its high stability and accuracy. Moreover, we perform temporal smoothing on the frame-wise SMPL parameters and obtain the positions of the 3D keypoints $J \in \mathbb{R}^{24 \times 3}$ through motion retargeting. Finally, we

Table 3: Quantitative comparison with state-of-the-art models. Our model is evaluated against both leading closed-source models and open-source baselines. **Bold** indicates the best performance and second-best results are underlined.

| | Auto Metrics | | | | Human Eval | | |
|---|---|---|---|---|---|---|---|
| | Human Anatomy | Motion Smoothness | Dynamic Degree | Aesthetic Quality | Video Quality | Prompt Following | Posture Plausibility |
| Veo 3.1 | **85.5** | **99.3** | 67.5 | **59.9** | 90.2 | **89.9** | **92.1** |
| Kling 2.5 Turbo | 84.7 | 99.0 | **70.0** | 59.7 | **90.4** | 89.3 | 91.0 |
| Wan-5B | 82.3 | 98.7 | 62.2 | 58.3 | 72.7 | 77.5 | 69.1 |
| EchoMotion(Wan-5B) | 83.2 | 99.1 | 64.0 | 58.2 | 80.4 | 81.4 | 80.8 |

leverage Qwen-VL-Narrator (qwe) to caption the videos. Our construction mechanism yields about 80k high-quality paired datasets.

## A.5 COMPARISON WITH TOP-TIER COMMERCIAL MODELS.

We conduct a quantitative comparison with leading commercial models, evaluating on 40 prompts specifically selected for their motion complexity (Table 3). While a gap exists compared to top-tier closed-source models like Veo 3.1 (veo) and Kling 2.5 Turbo (kli), this is an expected outcome given the vast disparities in model scale, training data, and computational resources. Despite this, our results highlight two key takeaways: First, EchoMotion achieves a competitive Motion Smoothness score, confirming the effectiveness of our joint modeling approach for kinematic quality. Second, it significantly outperforms its video-only baseline (Wan-5B), especially in Posture Plausibility and Human Anatomy. This empirically validates that our method provides substantial gains in human video generation.

## A.6 MORE QUALITATIVE RESULTS

In this section, we provide an expanded gallery of qualitative results to further demonstrate the capabilities of EchoMotion.

Figure 10 showcases a diverse set of text-to-video generation results, highlighting the model's ability to handle complex prompts across various activities and environments. Figure 11 focuses on the motion-to-video task, illustrating how EchoMotion precisely translates 3D motion sequences into realistic videos while adhering to textual descriptions of appearance and context.

## A.7 MORE QUANTITATIVE RESULTS

### A.7.1 TEXT-TO-VIDEO.

To provide a more granular understanding of our model's performance and robustness, we present a detailed quantitative comparison across various motion categories. Figure 12 visualizes the performance of our proposed model, EchoMotion(Wan-5B), against several baselines and ablations. The evaluation is conducted on a diverse set of nine categories—Athletic, Arts, Ball, Daily, Fight, Gymnastics, Ice Show Sports, Outdoor, and Water Sports—in addition to an overall average performance. Models are assessed on four key metrics: Aesthetic Quality, Dynamic, Human Anatomy, and Motion Smoothness.

A consistent trend emerges from the results: while all models achieve competitive scores in Aesthetic Quality, our final model, EchoMotion(Wan-5B), establishes a significant lead in the metrics most critical to motion fidelity: Human Anatomy and Motion Smoothness. For instance, in the "Average Performance" comparison, our model outperforms the next-best baseline by a clear margin in these two areas. This strongly indicates that our approach is particularly effective at generating anatomically correct human figures and ensuring temporally coherent, smooth movements, which are common challenges in human video generation.

The strength of our model is further validated by its consistent high performance across a wide spectrum of motion types. From the intricate and precise movements in "Gymnastics" and "Arts" to the large-scale, dynamic actions in "Water Sports" and "Fight," our model consistently ranks first.

*"On a rushing river, the splashing milky-white water intertwines with the dark gray jagged rocks, as a female athlete smiles while kayaking…"*

*"A young smoothly dribbles in from the right side, controlling the ball rhythmically with his right hand, his body crouched like a poised cheetah; as he approaches the basket, he suddenly stops, and with explosive power from his legs, leaps into the air, raising the basketball above his head and dropping it into the hoop…..."*

*"A white woman occupies the center of the frame, her hands tightly gripping the horizontal bar, her arms extended straight, her body hanging naturally like a pendulum. Then, she elegantly bends and lifts her legs, with her toes pointed straight towards the ceiling, briefly hovering before gently falling like a feather…"*

*"A female skateboarder executed a difficult aerial flip in an outdoor U-shaped pool. She was dressed in a form-fitting black long-sleeve shirt, light blue jeans, and pure white sneakers, with her golden hair flowing in the wind. As she landed, her feet made solid contact with the slope, her knees bent flexibly to absorb the impact, and her arms extended like eagle wings to stabilize her center of gravity…."*

*"In a quiet indoor corner, the light blue walls radiate a warm blue tone under soft volumetric light, while the light-colored wooden floor reflects a delicate sheen. A middle-aged woman with curly hair and sunglasses walks in from the left side of the scene, then she bends down, placing her hands on a yoga mat, with her legs in a lunge position to stretch…"*

*"At the summit of a magnificent snow-covered mountain, a man dressed in a bright blue-green hooded ski suit launches himself into the air from a steep slope, smoothly executing an elegant backflip in mid-air, his snowboard tracing a dynamic arc as he spins; he lands by bending his knees to absorb the impact and continues to glide steadily…"*

*"... A young white male surfer stands barefoot on a light blue surfboard; his body is slightly bent like a bow, arms spread wide like wings to maintain balance with precision, eyes focused ahead, smoothly performing the difficult maneuver of shifting his body weight and rotating the board."*

*"At the golden moment on a professional athletic field, a strong young male athlete with a deep skin tone and neat black short hair is running steadily from the left side of the frame to the right, his body leaning forward with a sense of power, arms swinging rhythmically, steps coordinated and powerful, with a focused face full of training passion…"*

*"On the deep gray rubber mat of a modern gym, a woman in bright blue sports underwear and high-waisted tight shorts is fully focused on rope training. She leans forward, her back straight as a blade, legs slightly bent to gather strength, gripping the black padded handles firmly, pulling the rope down to her chest in a smooth and controlled rhythm, then slowly returning it to the starting point…"*

*"In the kitchen, the sunlight outlines the plump contours of the bright red tomatoes on the wooden cutting board. A woman with a sincere smile is facing the camera, her hands gracefully raised to her chest with palms facing outward to show an open posture. She then naturally brings her hands together in front of her chest, maintaining a friendly expression, and slightly bows to convey a naturally warm friendliness..."*

Figure 10: Additional text-to-video generation results from EchoMotion. These examples showcase the model's capability to generate a diverse range of high-quality videos, spanning various activities (e.g., kayaking, skateboarding, skiing) and environments. The results demonstrate a faithful adherence to complex and detailed textual descriptions.

This demonstrates the generalizability and robustness of our method, proving it is not overfit to a specific type of motion but can handle a diverse range of human activities effectively.

### A.7.2 MOTION-TO-VIDEO.

We provide a quantitative comparison on the motion-to-video task against several leading methods on the case from VACE-Benchmark (Jiang et al., 2025), including Text2Video-Zero (Khachatryan et al., 2023), Follow-Your-Pose (Ma et al., 2024a), VACE-14B (Jiang et al., 2025), and the specialized animation model Wan2.2-Animate-14B (Cheng et al., 2025). As shown in Table 4, EchoMotion achieves highly competitive results, demonstrating excellent video quality and strong kinematic plausibility with qualitative results provided in Figure 4. As shown in Table 4, EchoMotion achieves highly competitive results, demonstrating excellent video quality and strong kinematic plausibility. The qualitative results in Figure 13 visually corroborate these metrics, showcasing the model's ability to generate high-fidelity videos where the character's movement is highly faithful to the input motion, while the character's appearance and the background scenery strictly adhere to the text prompt.

It is important to clarify that our task is a cross-modal completion challenge (generating from both text and motion), which is distinct from standard image animation (driving a source image with mo-

*"An Asian young man is standing in the center of a dimly lit gym, wearing a dark gray hoodie and black tight training pants, engaged in strength training. The background features neatly arranged dark metal racks, which hold dumbbells and colored solid balls—orange, deep blue, and dark green spheres are interspersed, with a clearly visible brick wall texture in the back...."*

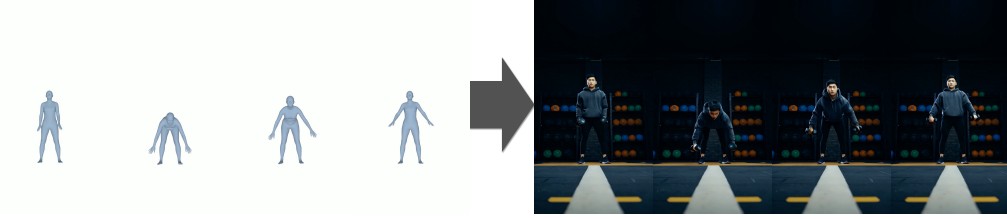

*"A young South Asian male is doing physical training on a rural path at dusk. He has a lean and muscular build, short hair, and a light beard. He is wearing a dark gray sleeveless sports vest and navy blue shorts, decorated with light gray stripes at the hem. He is barefoot on the dirt path. The background features an open, fallow farmland, with dried straw scattered on the ground...."*

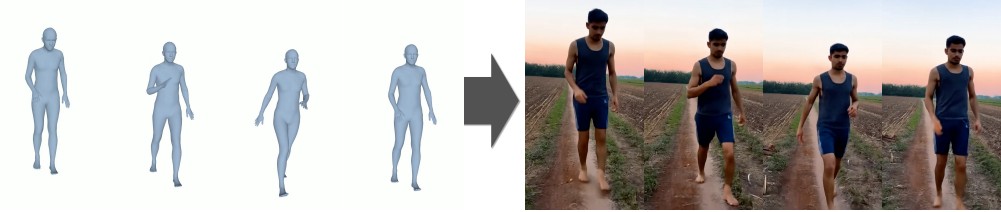

*"A young Asian woman is dressed in a dark turtleneck knit sweater and loose-fitting trousers, set in a minimalist industrial-style indoor space. The background features a textured concrete wall and a large floor-to-ceiling window. The shot is a medium shot, stable and fixed, with black-and-white tones maintaining the original high-contrast style, and the light and shadow structures remain unchanged, creating a calm yet tense overall atmosphere...."*

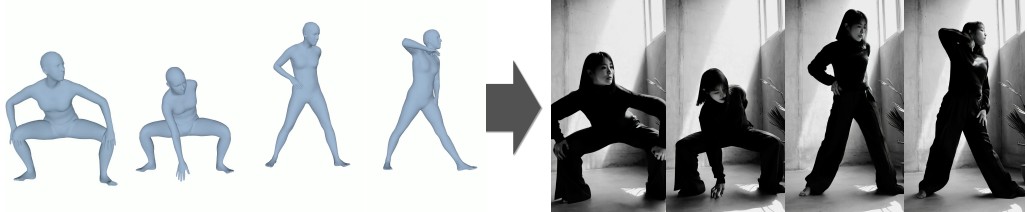

*"A deep-skinned, athletic African female athlete is in the final phase of her run-up for long jump at an outdoor track and field venue. She is on a deep red rubber track, with light yellow marker blocks visible at the edge of the track. The background features slightly blurred palm trees and shrubs, interspersed with scattered orange tropical flowers, and in the distance, there is the silhouette of low gray stands......"*

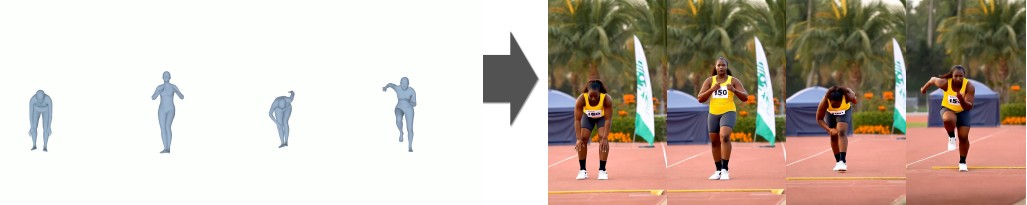

Figure 11: Additional results for the motion-to-video task. These examples illustrate EchoMotion's ability to accurately render a given 3D motion sequence into a visually coherent video. Crucially, the model follows the kinematic guidance from the motion input for the action, while simultaneously adhering to the textual prompt for the subject's appearance, attire, and scene context.

tion). For the comparison with Wan2.2-Animate-14B, which requires a source image, we provided the first frame of the ground truth video and transformed it by Flux-Kontext (Batifol et al., 2025) as its input. Since this setup does not use a text prompt to control for the Wan2.2-Animate-14B model, the Prompt Following metric is not applicable for this method.

The result highlights a key advantage of EchoMotion: its ability to jointly understand and synthesize from both textual and motion inputs, offering greater flexibility. It is particularly noteworthy that despite being a more compact (5B) and versatile motion-video multi-task model, EchoMotion delivers performance on par with larger models specialized for a single task. Crucially, this versatility is achieved with remarkable parameter efficiency. Unlike approaches that require training an auxiliary control module (e.g., a ControlNet branch), EchoMotion natively supports the motion-to-video task through its unified motion-video multi-task training, without introducing any additional parameters.

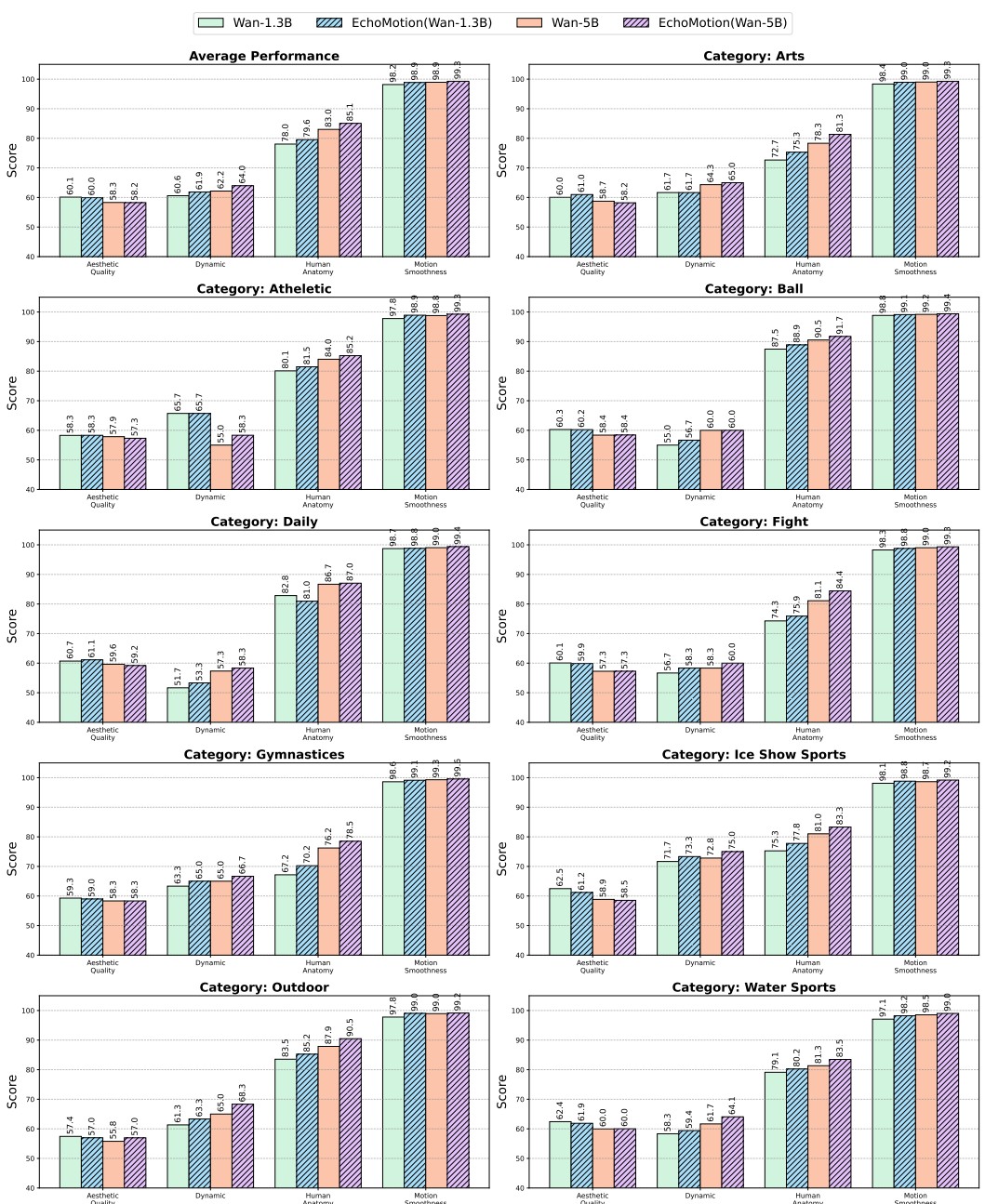

Figure 12: Per-category quantitative comparison. We provide a detailed performance breakdown of our model (Wan2.2 5B + EchoMotion) and other baselines across nine distinct motion categories, plus an overall average. Models are evaluated on four metrics: Aesthetic Quality, Dynamic, Human Anatomy, and Motion Smoothness. Our model consistently achieves superior performance, particularly in Human Anatomy and Motion Smoothness, across all tested scenarios, demonstrating its robustness and high-fidelity motion generation capabilities.

### A.7.3 VIDEO-TO-MOTION.

Following the evaluation protocol of recent works like ChatHuman (Lin et al., 2025), we assess the performance of our Video-to-Motion capability on 200 samples from the 3DPW test set (Von Marcard et al., 2018). We report the Mean Per-Joint Position Error (MPJPE) and Procrustes-Aligned

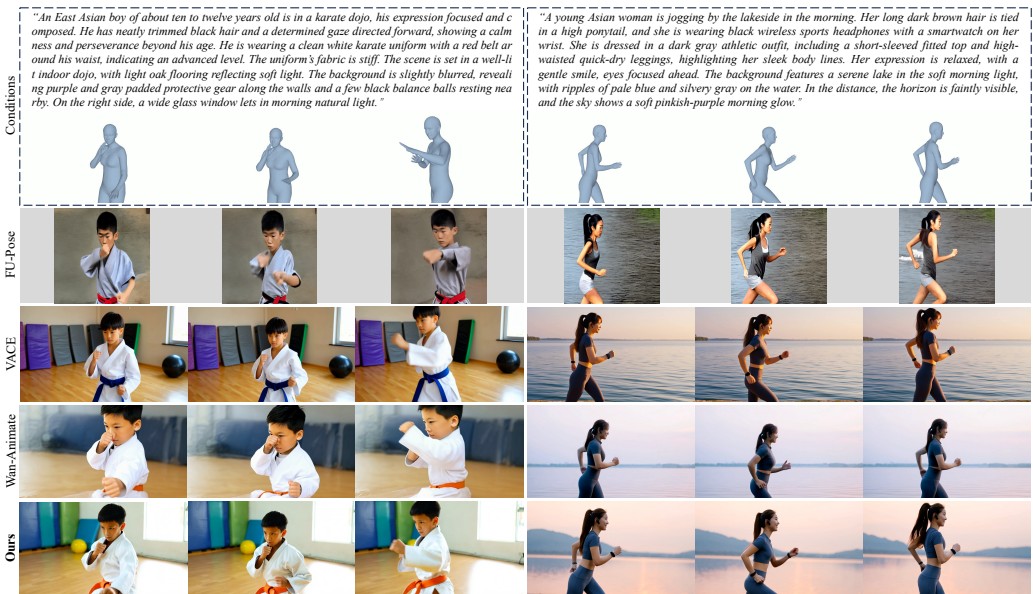

Figure 13: Qualitative comparison with the Motion-to-Video task with baseline methods.

Table 4: Quantitative comparison on human motion generation. Best results are highlighted in **bold** and second-best results are underlined.

| | Video Quality | | | | User Study | | |
|---|---|---|---|---|---|---|---|
| | Aesthetic Quality | Motion Smoothness | Overall Consistency | Temporal Flickering | Prompt Following | Pose Consistency | Overall Quality |
| Text2Video-Zero | 57.6 | 79.7 | 23.9 | 76.6 | 44.2 | 70.1 | 40.9 |
| Follow-Your-Pose | 48.8 | 90.1 | 26.1 | 88.0 | 50.9 | 79.2 | 42.1 |
| VACE-14B | 60.2 | 98.6 | 26.4 | 97.3 | **79.2** | 82.4 | 66.9 |
| Wan2.2-Animate-14B | **61.4** | 98.9 | **28.1** | **99.2** | - | **87.2** | **68.9** |
| EchoMotion(Wan-5B) | 59.2 | **99.1** | 26.9 | 98.4 | 78.2 | 82.2 | 65.2 |

MPJPE (PA-MPJPE) and categorize the compared methods into two groups: reconstruction-based and generative-based, as shown in Table 5.

The results show that while our method does not surpass specialized, reconstruction-based models like HMR2.0Goel et al. (2023), this is an expected outcome. Reconstruction-based methods are typically optimized for the single task of human mesh recovery and often rely on strong supervision from 2D keypoint annotations during training. In contrast, our EchoMotion model is designed as a versatile, motion-video multi-task generative framework, which inherently involves a trade-off between specialization and flexibility. Within the more comparable category of generative-based approaches, our method achieves performance on par with the state-of-the-art ChatHuman, demonstrating the effectiveness of our unified model, which supports not only motion recovery but also broader generative tasks.

### A.7.4 MOTION QUALITY.

While EchoMotion is primarily designed for video generation, we also evaluate its motion synthesis quality. Table 6 compares its performance with that of specialist motion generation models. Notably, a direct comparison using the Frechet Inception Distance (FID) with baseline methods is not applicable, as this metric is highly dependent on the training data distribution. Our model is trained

Table 5: Quantitative comparison of Human Mesh Recovery performance on the 3DPW dataset (Von Marcard et al., 2018). We categorize methods into reconstruction-based and generative-based approaches. Best results are highlighted in **bold**.

|  | Method | PA-MPJPE ↓ | MPJPE ↓ |
|---|---|---|---|
| Rec-Based | SPIN (Kolotouros et al., 2019) | 62.9 | 102.9 |
|  | HMR2.0 (Goel et al., 2023) | **58.4** | **91.0** |
| Gen-Based | ChatPose (Feng et al., 2024) | 81.9 | 163.6 |
|  | ChatHuman (Lin et al., 2025) | 58.7 | 91.3 |
|  | EchoMotion(Video-to-Motion) | 59.8 | 94.1 |

Table 6: Quantitative comparison of motion generation quality. The metrics are: Frechet Inception Distance (**FID**), Pose Plausibility (**PP**), Prompt Following (**PF**), and Motion Smoothness (**MS**).

| Method | FID ↓ | PP ↑ | PF ↑ | MS ↑ |
|---|---|---|---|---|
| MLD | - | 71.4 | 63.7 | 91.5 |
| MotionGPT | - | 80.1 | 72.3 | **93.9** |
| EchoMotion (Wan 1.3B) | **10.9** | 79.5 | 78.8 | 92.2 |
| EchoMotion (Wan 5B) | 11.3 | **80.8** | **82.3** | 92.4 |

on our proposed *HuMoVe* dataset, whereas the baseline models (*e.g.*, MLD (Chen et al., 2023), MotionGPT (Jiang et al., 2023)) are trained on distinct, motion-only datasets like HumanML3D (Guo et al., 2022). Consequently, we report FID scores only for the model trained on *HuMoVe* dataset.

For a fair and practical comparison, we generated a set of 50 diverse prompts using an LLM (Team et al., 2024), rendered the generated motion parameters into mesh videos, and then asked human annotators to score them on three key aspects: Pose Plausibility (PP), Prompt Following (PF), and Motion Smoothness (MS). The results show that EchoMotion achieves competitive performance on Pose Plausibility and Motion Smoothness, and excels in Prompt Following. We attribute this superior instruction-following ability to our joint training and sampling paradigm, where the motion module benefits from the rich semantic understanding of the large pre-trained video model.

## A.8 MORE ABLATION STUDIES

In this section, we present additional ablation studies to further validate the architectural design choices of EchoMotion.

### A.8.1 IMPACT OF POSITIONAL COLLISIONS IN MVS-RoPE

To demonstrate the necessity of the spatial extension design within our proposed MVS-RoPE, we investigate the effect of spatial index overlap between modalities. We compare two distinct configurations: (a) EchoMotion (Proposed): Motion tokens are treated as a diagonal extension of the visual latent space to ensure unique spatial identifiers. The encoding is formulated as:

$$\hat{\boldsymbol{f}}_{t,i}^m = \text{MVS-RoPE}(\boldsymbol{f}_{t,i}^m, t, i) = \mathcal{R}(t, H + i, W + i) \cdot \boldsymbol{f}_{t,i}^m, \tag{14}$$

where $H$ and $W$ denote the spatial dimensions of the video latents.

(b) Positional Collision (Baseline): Motion tokens are assigned spatial indices starting from the origin, identical to the initial visual tokens. This results in a "collision" of coordinates:

$$\hat{\boldsymbol{f}}_{t,i}^m = \text{MVS-RoPE}(\boldsymbol{f}_{t,i}^m, t, i) = \mathcal{R}(t, i, i) \cdot \boldsymbol{f}_{t,i}^m. \tag{15}$$

Both models were trained for 1,600 iterations under identical settings. The qualitative comparison is illustrated in Figure 14. The results reveal a stark contrast: the model utilizing our proposed MVS-RoPE (Left) produces visually appealing and temporally coherent videos. Conversely, the Positional

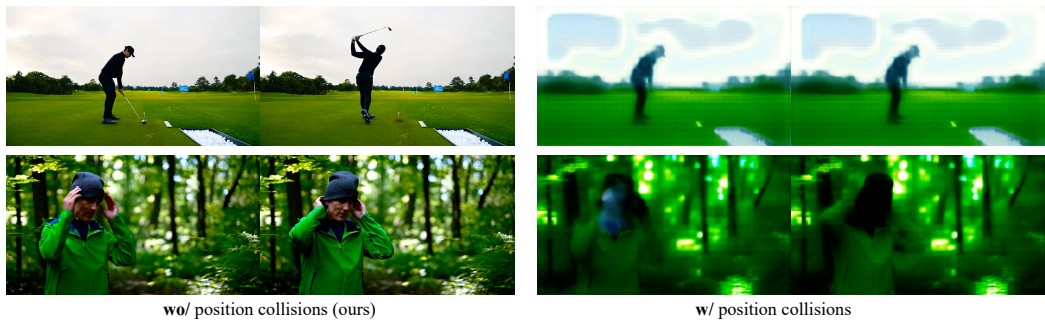

**wo/** position collisions (ours)         **w/** position collisions

Figure 14: The negative effect of positional collision of the positional embedding.

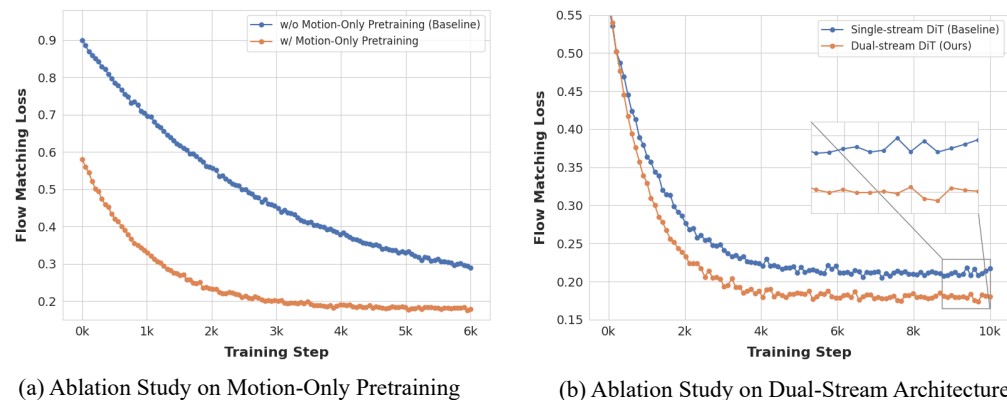

(a) Ablation Study on Motion-Only Pretraining      (b) Ablation Study on Dual-Stream Architecture

Figure 15: Ablation studies on our key designs. (a) Comparison of training loss between our two-stage training scheme (w/ Motion-Only Pretraining) and a one-stage baseline (w/o Motion-Only Pretraining). (b) Comparison of training performance between our Dual-stream DiT and a Single-stream DiT baseline. Our architecture achieves a consistently lower loss and converges to a better final value.

Collision baseline (Right) suffers from severe visual degeneration, generating output with collapsed structures and noise.

This degradation occurs because identifying the modality of a token relies heavily on its positional embedding in the attention mechanism. When motion and video tokens share the same spatial coordinates (collision), it creates positional ambiguity. This forces the motion tokens to interfere with the visual tokens in the attention mechanism, disrupting the pre-trained generative priors of the video backbone and preventing the model from effectively distinguishing between the latent representations of the two modalities.

### A.8.2 TWO-STAGE TRAINING SCHEME.

To validate the effectiveness of our motion-only pretraining, we conduct an ablation study comparing the Phase 2 (motion-video joint training) convergence of two models. Our full model starts this phase from the checkpoint obtained after motion-only pretraining, while the baseline model skips Phase 1 and begins joint training with its motion branch initialized by copying the weights from the video branch. As illustrated in Figure 15(a), the benefits are clear and significant. Our model with pretraining (orange curve) not only (1) begins with a substantially lower Flow Matching Loss, but also (2) converges much more rapidly and (3) achieves a better final loss value compared to the baseline (blue curve). This demonstrates that establishing a strong motion prior in Phase 1 provides a superior initialization for the subsequent joint training. This pretraining prevents the motion branch's learning signals from being overwhelmed by the computationally dominant video branch, leading to a more stable and efficient joint distribution learning.

### A.8.3 DUAL-BRANCH ARCHETECTURE.

We conduct an ablation study to verify the superiority of our dual-branch DiT architecture over a single-stream alternative. The single-stream baseline processes the concatenated video and motion tokens through a shared set of Q, K, V projection and FFN layers. In contrast, our dual-branch design employs a separate set of weights (i.e., experts) for the projection and FFN layers of each modality.

As illustrated in Figure 15(b), the training loss curves for both architectures reveal a clear performance gap. Our dual-branch model (orange curve) consistently achieves a lower flow matching loss throughout the training process compared to the single-stream baseline (blue curve). Notably, our model converges to a significantly lower final loss, while the baseline plateaus at a higher value, as highlighted in the magnified inset. This result suggests that providing dedicated processing paths for video and motion modalities allows the model to learn more effective and specialized representations. This approach avoids the potential feature interference that can occur in a shared-weight, single-stream architecture, ultimately leading to superior model performance and a more robust joint distribution model.

### A.9 EXPERIMENTS DETAILS

We adapt two pre-trained video foundation models, Wan2.1-1.3B and Wan2.2-5B, by integrating our dual-modality blocks.

- For the 1.3B model, we replace all original DiT blocks with our video-motion blocks. This results in a final model with 2.6B parameters.
- For the larger 5B model, we adopt a hybrid strategy, replacing half of the video blocks with our dual-modality blocks, which yields a final model with 7.5B parameters.

All models were trained on NVIDIA A100 80GB GPUs. The 2.6B model required approximately 2,300 GPU hours for pre-training, while the 7.5B model required 4,000 GPU hours. Detailed training hyperparameters are provided in Table 7.

Table 7: Experimental settings of EchoMotion 1.3B model.

| Parameter | Value |
|---|---|
| Transformer dim | 1536 |
| Numbers of heads | 24 |
| Numbers of layers | 30 |
| Number of video-only blocks | 0 |
| Number of video-motion blocks | 30 |
| Video height | 480 |
| Video width | 832 |
| Video frame | 81 |
| FPS | 16 |
| Batchsize | 2 |
| Train timesteps | 1000 |
| Train shift | 8.0 |
| Optimizer | AdamW |
| Learning rate | 8e-6 |
| Weight decay | 0.001 |
| Sample timesteps | 50 |
| Sample shift | 8.0 |
| Sample guidance scale | 6.0 |

### A.10 COMPUTATION AND PARAMETER ANALYSIS

As illustrated in Figure 16, it is crucial to distinguish between our model's parameter count and its computational cost. While introducing the dual-branch architecture significantly increases the total number of parameters, the growth in computational overhead (i.e., FLOPs) is disproportionately

Table 8: Experimental settings of EchoMotion 5B model.

| Parameter | Value |
|---|---|
| Transformer dim | 3072 |
| Numbers of heads | 24 |
| Numbers of layers | 30 |
| Number of video-only blocks | 15 |
| Number of video-motion blocks | 15 |
| Video height | 708 |
| Video width | 1280 |
| Video frame | 121 |
| FPS | 24 |
| Batchsize | 1 |
| Train timesteps | 1000 |
| Train shift | 8.0 |
| Optimizer | AdamW |
| Learning rate | 8e-6 |
| Weight decay | 0.001 |
| Sample timesteps | 50 |
| Sample shift | 8.0 |
| Sample guidance scale | 6.0 |

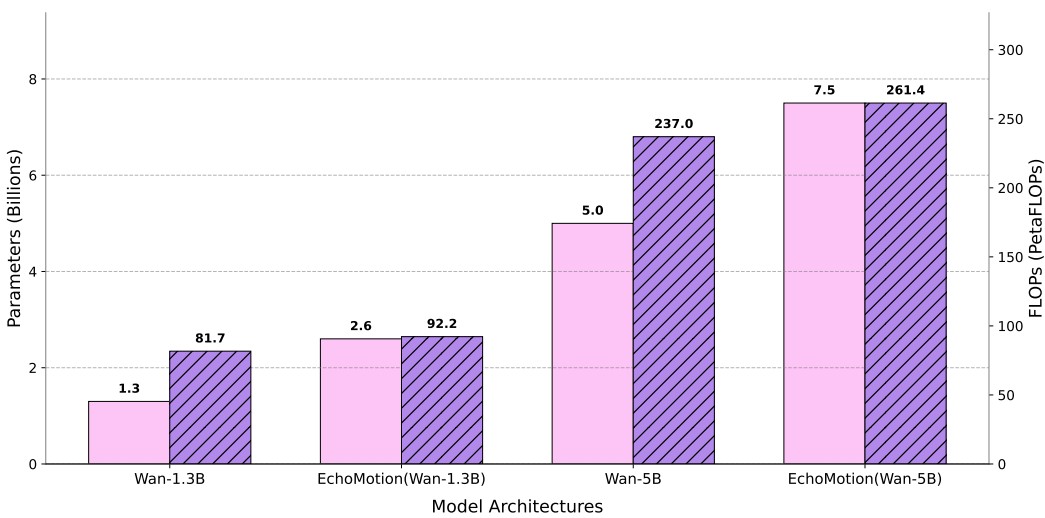

Figure 16: Visualization of our model scale and computational cost.

small. For instance, our 1.3B dual-branch model has double parameters than its video-only counterpart but requires only 12.9% more FLOPs for a single forward pass. This trend holds true for our larger 5B models: the hybrid architecture introduces 51% more parameters but only increases the computational cost by 10.3% (from 237.0 to 261.4 PFLOPs).

This disproportionately small increase in FLOPs is a direct consequence of our architectural design. The additional parameters are exclusively dedicated to processing motion tokens, which, despite their rich information content, represent a small fraction of the total sequence length (e.g., 4,131 motion tokens vs. 32,760 video tokens in the 1.3B model). As a result, the impact on practical inference latency is marginal. This demonstrates that our dual-branch approach is a highly efficient method for incorporating multimodal capabilities, significantly expanding model functionality with a minimal increase in its computational footprint.

## A.11 SOCIETAL IMPACTS AND SAFEGUARDS

The advancements in controllable human video generation grounded in kinematic models, as exemplified by our work on EchoMotion, present profound societal impacts. By enabling the creation of high-quality, kinematically plausible videos with direct control over human motion, our framework democratizes access to sophisticated animation and visual effects tools. This innovation can dramatically streamline workflows in industries such as filmmaking (for pre-visualization), video games (for realistic character animation), virtual reality (for lifelike avatars), and even in specialized fields like sports science and physical therapy for visualizing complex biomechanics. The ability to generate videos from structured motion data offers unprecedented creative flexibility and a new paradigm for digital human creation.

However, the power of such generative technologies also introduces significant challenges. The automation of high-fidelity animation could lead to job displacement for traditional 3D animators and motion capture artists, necessitating industry-wide adaptation and a focus on creative direction over manual execution. More critically, the potential for misuse poses a serious ethical concern. The ability to generate realistic videos of individuals performing complex actions they never took could be exploited to create highly convincing deepfakes, impersonations, or misinformation, thereby eroding public trust. Furthermore, ensuring that our $MoVe$ dataset and the models trained on it are free from demographic or physical biases is essential to prevent the generation of content that reinforces harmful stereotypes.

To address these issues, we implement robust safeguards. Our models inherit the safety mechanisms from their foundational bases (Wan2.1 and Wan2.2), which include filters to detect and prevent the generation of inappropriate or harmful content. We are committed to adhering to strict ethical guidelines regarding the use of our technology. By open-sourcing our models and the $MoVe$ dataset, we aim to foster transparency, enable independent auditing, and encourage a community-driven approach to developing responsible and ethical human-centric generative AI.

