# OpenReview forum: "EchoMotion: Unified Human Video and Motion Generation via Dual-Modality Diffusion Transformer"
_ICLR.cc/2026/Conference — ICLR 2026 Poster_

### Official Review · Reviewer_GwaP · 2025-10-30

**Soundness:** 1
**Presentation:** 2
**Contribution:** 2
**Rating:** 4
**Confidence:** 4

**Summary:**

Inspired by VideoJam , this work establishes a model for the joint distribution of video and motion. It explicitly denoises parametric motion and performs text-to-video & motion generation. The results demonstrate an improvement in motion smoothness and human evaluation scores compared to the baseline (Wan video ).

**Strengths:**

Originality:

It designs and establishes a modeling framework for the joint distribution of video and motion.

Quality:

The quality is acceptable.

Clarity:

The paper is well-structured and clearly articulated.

Significance:
1. This work proposes a solution for modeling the joint distribution of video and motion.
2. Community Contribution: The authors commit to open-sourcing their code, which will be a valuable public resource for advancing the field

**Weaknesses:**

1. Limited quantitative experiments: The paper only compares results with its base model using metrics that are not specialized for human motion. It lacks comparisons with closed-source models like Kling or Veo3 (it doesn't necessarily need to surpass them, but at least show the gap with SoTA models). The evaluation metrics are not focused on human motion.



2. Lack of necessary ablation studies: The effectiveness of the video-to-motion and motion-to-video  capabilities is unknown, as no quantitative results are provided. This is crucial for validating the joint distribution modeling. Furthermore, there is no ablation study for the complex training process .


3. The visual quality demonstrated in the supplementary materials is still subpar. There are instances of impossible human poses, and the characters' hands are very blurry.

**Questions:**

1. In the supplementary materials, specifically in sample 6 (especially the last frame) and sample 15, some very unnatural or incomprehensible human poses appear. What are the possible reasons for this?

2. As mentioned in the paper, EchoMotion can perform video-to-motion and motion-to-video  tasks. Could you provide quantitative metrics to demonstrate the performance of these tasks? Specifically, for motion-to-video, could you compare it with models like Champ , Animate Anyone, or WanAnimate (since its base model is also Wan video)?



3. The VBench metrics  used in the comparison are not specifically designed for human motion. Would it be possible to compute an FID (Fréchet Inception Distance) on the generated SMPL motion parameters?

4. It is suggested to also include comparisons with closed-source models, such as Kling, Veo3, etc.

5. If you were to use SMPL-X as the motion representation instead of SMPL, would this lead to an improvement in the representation of hands?


6. Could an ablation study be provided for the complex training process described in Section 3.2 ?

---

> ### Author Response · Authors · 2025-11-21
> **Response to Review 4 Weakness**
>
> Thank you for your constructive feedback and recognition. Below are our responses, which we hope will address your concerns.
>
> > **Response to Weakness 1:**
>
> We thank the reviewer for this feedback on our experimental scope.
>
> We would like to clarify our paper's core contribution. Our central hypothesis is that the joint modeling of video and motion can significantly enhance the quality of human-centric video generation. Our primary experiments in Table 1, which compare EchoMotion against its strong base models (Wan-1.3B and Wan-5B), are designed to test this specific hypothesis. The results, showing clear improvements in human-focused metrics, provide a self-contained and rigorous validation of our method's effectiveness. We believe this contribution is significant, independent of comparisons to top-tier, closed-source models.
>
> To show the gap with SoTA models, we are actively conducting the experiments for the comparison with top commercial models. We plan to finalize this analysis within the next four days and will update this document with the full results.
>
> > **Response to Weakness 2:**
>
> We thank the reviewer for these valid points on the experimental validation.
>
> First,  we agree that quantitative validation of these individual tasks is crucial. As a follow-up to this feedback, we have conducted new experiments to provide a quantitative analysis of both the Motion-to-Video and Video-to-Motion modes, which can be seen in Sec. A.7.2 and Sec. A.7.3.
>
> Second, regarding the training process, we agree that an ablation study is necessary. We have provided a new ablation in the Sec A.8.2 to validate the effectiveness of our proposed two-stage training strategy. Specifically, our model with motion pretraining reaches a low motion loss convergence threshold more than 20 times faster in terms of GPU hours compared to the baseline that trains both modalities jointly from the beginning.
>
>
> > **Response to Weakness 3:**
>
> We thank the reviewer for this feedback on the qualitative results.
>
> First, regarding "impossible human poses," we acknowledge that generating perfect human anatomy in all scenarios remains a highly challenging open problem for all generative models. Our core contribution is not that we have solved this entirely, but that our joint modeling of kinematics significantly reduces these artifacts. This is validated by our quantitative improvements in "Human Anatomy" and "Posture Plausibility" (Table 1), and our qualitative comparisons (Figure 5).
>
> Second, regarding "blurry hands," this is a known challenge for most video models, including SoTA, and is often related to resolution and the VAE's capabilities. As discussed in Reviewer 2 Weakness 2 response, our SMPL-based motion branch does not provide constraints for hands, so their fidelity is determined by the pre-trained visual branch. Our work focuses on improving the body's kinematic structure and hopes to solve this problem in the future.

---

> > ### Comment · Reviewer_GwaP · 2025-11-25
> >
> > I thank the authors for their response and additional experiments. I acknowledge that the proposed method performs well on both motion-to-video and video-to-motion tasks.
> >
> > I still have some questions:
> > - The training strategy remains confusing to me. I suggest the authors clarify the organization of the main paper regarding the following points: How many training steps are required for the motion-only pretraining phase? How many training steps are needed for Phase 2? The description of "multi-task" in Phase 2 appears too broad. How do the authors determine the stopping criteria for Phase 1 and Phase 2?
> >
> > - Furthermore, Figure 14(a) is not convincing. Since the y-axis represents the motion flow matching loss, it is naturally expected that a model with motion pretraining would exhibit a faster decline in loss.
> >
> > - Finally, could you provide side-by-side video comparisons with the baselines?
> >
> > If the authors can address these concerns, I will consider raising my score.

---

> > > ### Author Response · Authors · 2025-11-26
> > > **Response to Remaining Concerns from Reviewer 4**
> > >
> > > Dear Reviewer,
> > >
> > > We sincerely thank you for your constructive feedback. Your insightful questions are instrumental in helping us further refine and clarify our manuscript, and we are grateful for the opportunity to address them. We have uploaded a revised version of our paper that incorporates these revisions, and we detail our responses below. We believe these changes and clarifications will fully address your concerns.
> > >
> > > > ## Respond to Concern 1:
> > >
> > > **Training Steps:** The Phase 1 (Motion-only Pretraining) is conducted for 20k steps, and Phase 2 (Motion-Video Multi-task Training) for an additional 12k steps. We now explicitly state this detail in Section 4.1 in the revised manuscript.
> > >
> > > **Stopping Criteria:** The training step counts were determined by monitoring both loss convergence and qualitative generation quality on a held-out validation set. We have added a formal description of our stopping criteria in Section 4.1 in the revised paper.
> > >
> > > **The description of "multi-task" training:** We agree that "multi-task" was too broad. We have refined this term to "motion-video multi-task training" throughout the paper. To further clarify, we now explain in the caption of Figure 3 that we regard "motion-with-video", "motion-to-video", and "video-to-motion" as three distinct tasks and conduct mixed training on them. This refinement simultaneously reflects the characteristics of both multimodal and multi-task training. We believe this revised term is more precise, but we would be happy to make further revisions based on your feedback.
> > >
> > > > ## Respond to Concern 2:
> > >
> > > Thank you for your careful observation. We agree that a faster decline is expected in Figure 14 (Figure 15 in our revised paper). We believe the reasons are as follows:
> > >
> > > **Reason 1: Logarithmic Scale for X-axis:**  The x-axis is plotted on a log scale to better visualize the performance gap during the initial training phase. While this makes the curve appear less steep, it is a standard visualization technique for comparing convergence rates over orders of magnitude. The trade-off for this enhanced early-stage visibility is that the descent appears more gradual compared to how it would on a linear-scale plot.
> > >
> > > **Reason 2: Learning Rate Warm-up Schedule:** To preserve the powerful priors from the pre-trained DiT, we employ a conservative learning rate warm-up schedule (for both settings in this ablation study). Specifically, the learning rate linearly increases from 0 to 1e-5 for the first 2k training steps. This intentional design choice prioritizes training stability over a rapid initial loss decrease.
> > >
> > > > ## Respond to Concern 3:
> > >
> > > We thank the reviewer for this suggestion. We have provided side-by-side video comparisons with baselines in Figure 13 and disscusion in Section A.7.2 in the revised paper.
> > >
> > > We hope that these substantial revisions and clarifications will address your concerns. Thank you once again for your valuable guidance in strengthening our work.

---

> > > > ### Comment · Reviewer_GwaP · 2025-11-27
> > > >
> > > > > Question 2:
> > > >
> > > > I feel that my question has not been addressed. My concern is that the rapid decline in motion flow matching loss does not verify the superiority of the 'motion pretraining first' strategy. Given that the red curve represents the model with motion pretraining, it is naturally expected to exhibit faster loss convergence in subsequent training phases.
> > > >
> > > > > Question 3:
> > > >
> > > > The superiority of the proposed method is not clearly observable from the figures provided. To convincingly demonstrate the improvements, I strongly suggest updating the response with video comparisons (dynamic results).

---

> ### Author Response · Authors · 2025-11-21
> **Response to Review 4 Question**
>
> > **Response to Question 1:**
>
> Please see our response to Reviewer 4, weakness 3.
>
> > **Response to Question 2:**
>
> To validate the performance on the cross-modal completion task, we have conducted the new quantitative experiments that the reviewer suggested and included them in the final manuscript. Specifically:
>
> * **Evaluating the Motion-to-Video task:** We provided a quantitative comparison of our model (in M2V mode) against specialized conditional generation methods, using Ground Truth (GT) motion as the input condition. Please see Sec.  (please see Sec. A.7.2)
> * **Evaluating the Video-to-Motion task:** We will also quantitatively evaluate our model's Video-to-Motion (inverse kinematics) capability, comparing its motion recovery accuracy against baselines.  (please see Sec. A.7.3)
>
> We agree this will "better highlight the advantage of joint modeling" (as the reviewer astutely noted), even if our unified model does not surpass every specialized model on every task. These new ablations will provide a much clearer picture of the benefits of our unified approach.
>
> > **Response to Question 3:**
>
> We thank the reviewer for this insightful comment on our evaluation metrics.
>
> This is an insightful suggestion.  We are actively conducting the experiments for the motion generation results. We plan to finalize this analysis within the next four days and will update this document with the full results.
>
> > **Response to Question 4:**
>
> Please see our response to weakness 1.
>
> > **Response to Question 5:**
>
> That is an excellent question. We view the use of SMPL-X as a necessary, but not sufficient, condition for improving hand representation. Our early experiments showed that its effectiveness is currently hindered by other factors, primarily:
>
> - The **hand pose annotations** in our dataset were not consistently accurate, providing a noisy signal for the model.
> - The **text captions** often lacked the specific, fine-grained details required to supervise complex hand gestures.
> We plan to address this in future work, as it is a non-trivial challenge.
>
> > **Response to Question 6:**
>
> Please see our response to Reviewer 4 Weakness 3 and Section. A.8.2.

---

> ### Author Response · Authors · 2025-11-25
> **Update: Comparison with Top-Tier Closed-Source Models and Quantitative Results on Motion Quality Now Included**
>
> Thank you once again for your detailed and constructive feedback. We have uploaded a revised version of our manuscript that incorporates new experimental results. We hope these substantial additions and clarifications address your concerns.
>
> > **Update Results for Weakness 1 and Question 4.**
>
> Thank you for your constructive feedback regarding the scope of our experiments. We appreciate the suggestions and have diligently worked to address your concerns in the revised manuscript. As per your recommendation, we have conducted an extensive quantitative comparison with leading closed-source models (Veo 3.1 and Kling 2.5 Turbo) to contextualize our method's performance and "show the gap." These new results are now included in Section A.5 of the Appendix (Table 3). We believe the newly added SOTA comparisons and the clarification of our human-focused metrics provide a much more comprehensive and rigorous evaluation of our work.
>
> > **Update Results for Question 3.**
>
> While our paper's primary focus is on 1) how joint video-motion modeling enhances video quality, and 2) how these two distinctive modalities control each other by cross-modality completion, the revised manuscript also includes a quantitative analysis of the underlying human motion synthesis (Appendix A.7.4, Table 6). As requested, we computed the FID on motion parameters on the *HuMoVe* dataset. However, due to differing training datasets between our model (HuMoVe) and the baselines (HumanML3D), a direct FID comparison is not applicable. We therefore report our FID scores across different scales for reproducibility. To enable a fair comparison, we then conducted a human evaluation by rendering all generated motions into mesh videos (please see Table 6 in the Appendix).

---

> ### Author Response · Authors · 2025-11-27
> **Response to Additional Question from Reviewer 4**
>
> Thank you very much for your insightful follow-up questions and constructive suggestions. We sincerely appreciate the time and effort you have dedicated to helping us improve our work.  We have made substantial revisions to our manuscript and supplementary materials.
>
> > ## Response to Question 2:
>
> We appreciate you pointing this out and agree that our previous figure could be misinterpreted as merely an expected outcome of pretraining. To provide a more direct and compelling ablation study, we have replaced the original figure. The new figure compares two settings for the Phase 2 training: (1) Our full model, which starts from the checkpoint obtained after motion-only pretraining, and (2) a baseline model, which skips phase 1 and begins phase 2 training with its motion branch initialized by copying the weights from the video branch.
>
> As the new figure illustrates, our model with pretraining (orange curve) not only starts with a lower loss but also demonstrates faster convergence to a lower final loss value, providing clear evidence for the effectiveness of our two-stage approach.
>
> > ## Response to Question 3:
>
> Thank you for your valuable feedback. We completely agree that dynamic video comparisons are essential for a convincing demonstration of our method's capabilities. In response, we have made the following clarifications and updates:
>
> First, we wish to clarify that our method operates at a smaller scale (in terms of data, parameters, and computational resources) compared to top-tier models like VACE-14B and Wan-2.2-Animate, which present an inherent challenge in direct comparisons. Nevertheless, our quantitative and qualitative results show that our approach serves as an effective unified framework for joint generation and cross-modal completion, achieving comparable performance to these state-of-the-art methods on the motion-to-video task. To provide this direct evidence, we have now uploaded new motion-to-video comparison videos to the supplementary materials. These videos temporarily replace the previous content due to space limitations.
>
> We hope that these substantial revisions and clarifications will address your concerns.

---

### Official Review · Reviewer_T7MT · 2025-10-31

**Soundness:** 3
**Presentation:** 3
**Contribution:** 2
**Rating:** 4
**Confidence:** 4

**Summary:**

This paper aims to overcome the limitations of motion generation based on pixel-level supervision in previous studies by proposing joint modeling of human appearance and motion.

The authors propose a DiT-based architecture that processes tokens from two modalities. The SMPL parameters are used to represent human poses, and to emphasize the dual-modality nature, Query, Key, and Value extracted from both the video and motion are concatenated and processed through self-attention. This structure enables attention to consider multiple modalities, which is advantageous for joint modeling. A motion-video synchronized RoPE (MVS-RoPE) which is an encoding method applicable to both modalities, is also proposed. Specifically, a diagonal extension is proposed to prevent interference between motion latents and video latents.

In addition, the authors propose the HuMoVe dataset, containing over 80,000 video-motion pairs. This dataset includes descriptive textual captions, 3D SMPL motion parameters, and video pairs, making it valuable for multi-modal generative modeling that considers vision, text, and motion jointly.

The experimental results present various metrics and human evaluations, showing performance improvements over baselines. Furthermore, ablation studies for each module are provided to analyze the effectiveness of the proposed methods.

**Strengths:**

- The paper proposes the large-scale HuMoVe dataset. Since the dataset includes test captions, videos, and motion parameter pairs, it is highly useful for multi-modal modeling tasks.

- MVS-RoPE that can be jointly applied to visual and motion embeddings is proposed. This encoding technique utilizes diagonal positioning to prevent interference between vision and motion latents, which is a reasonable approach (although more experimental evidence is needed to support this).

- The paper is easy to follow.

**Weaknesses:**

- The deep network structure is only a simple extension of existing networks. Except for MVS-RoPE, the network mainly uses self-attention on concatenated features for joint modeling, which is quite simple and straightforward. Discussion on whether other components could be improved to better support joint modeling would strengthen the paper.

- The quantitative evaluation relies only on self-evaluation. Even if direct comparison with prior studies is difficult, the paper should include analyses comparing the video and motion decoder performance improved from joint modeling with existing conditional generation methods (e.g., VideoJAM) to show the degree of improvement or equivalence.

- The explanation of how text descriptions were generated for the HuMoVe dataset needs to be clarified. In particular, since the initial data were created using an LLM, the paper should provide more detailed information about the prompts used.

**Questions:**

- p.2 L66: The authors mention that previous works are limited because, even with a 3D prior, supervision is applied after projecting it into 2D, which constrains accurate 3D (motion) generation. However, since the proposed method is also trained through a video diffusion process, hasn’t it still failed to overcome the problem of losing 3D information?

- p.4 L189: Motion tokens are generated as 51 dimension. What is the specific reason for this number?

- Motion tokens are added diagonally to visual tokens. Since maintaining temporal alignment is sufficient, there seems to be no strict reason for using the diagonal arrangement. Is there experimental evidence supporting the "positional collisions" mentioned in the text?

- Eq. 6: If the reviewer's understanding is correct, the last term must be u_{\theta}(\phi, m_{t}, \phi)

- Quantitative comparison is provided only as self-evaluation. Although direct quantitative comparison with previous studies may be difficult, joint modeling is expected to enhance the performance of both the video and motion decoders. Therefore, a quantitative comparison between the videos and motions generated by the proposed framework and those produced by conditional generation methods (e.g., VideoJAM), given GT as condition, could better highlight the advantage of joint modeling (even if the performance does not surpass that of conditional generation).


- Minor Comments:

-- p.5 L231: i is the motion token -> i is the motion token index ?

-- Fig.3: The distinction between "noisy" and "clean" is described only in text. It would be clearer and easier to understand if visual symbols were added to indicate the presence or absence of noise.

---

> ### Author Response · Authors · 2025-11-21
> **Response to Review 3 Weakness (Part 1)**
>
> Thank you for your constructive feedback and recognition. Below are our responses, which we hope will address your concerns.
>
> > **Response to Weakness 1: Discussion on Other Components and Clarification of Novelty.**
>
> We thank the reviewer for this comment. We respectfully argue that our architecturally simple approach is a **deliberate and effective design choice** for modeling heterogeneous modalities (pixels and parametric tokens).
>
> As the reviewer suggests, we considered alternative designs, but they have critical flaws that our method solves:
>
> * **Alternative 1: Cascaded Model (Motion Generator $\rightarrow$ ControlNet):** This approach fails because motion datasets are small, creating a **generalization bottleneck** for the entire pipeline. It also forces the video model to follow a low-fidelity 2D rendering, **destroying high-frequency details** like faces and clothing.
>
> * **Alternative 2: Joint 2D Map Generation (e.g., RGB + Depth):** This is also sub-optimal. It **discards 3D information** during occlusion. More importantly, it **loses the SMPL kinematic prior** entirely, as the model generates a raw depth map with no built-in anatomical constraints.
>
> **Why Our "Simple" Approach is Superior:**
>
> Our joint modeling framework directly avoids these issues.
>
> 1.  **Mutual Generalization:** The generalizable video branch helps the motion branch generalize during the forward propagation.
> 2.  **Detail Preservation:** By using *sequence-level* concatenation (not channel-level), motion acts as a **"soft constraint"** on the body's pose, guiding the kinematics *without* overriding the visual branch's ability to render fine details.
> 3.  **Kinematic Prior:** We model the **parametric SMPL tokens directly**, ensuring the strong anatomical prior is preserved.
>
> We hope this discussion clarifies that our design, while straightforward, is a novel and more effective solution than other, more complex alternatives. Discussions about novelty can also be found in our response to Reviewer 1's Weakness 1.
>
> > **Response to Weakness 2: Quantitative Evaluation on Cross-Modal Completion. **
>
> **1. On Comparison with SOTA Joint Generation Methods (e.g., VideoJAM)**
>
> The reviewer correctly identifies VideoJAM as a relevant method that also explores joint modeling. Unfortunately, **VideoJAM is not open-sourced**, which makes a direct, quantitative comparison infeasible at this time. Additionally, VideoJAM is focused on the text-to-video task, not the conditional generation task.
>
> However, we can provide a detailed discussion of the fundamental differences in our approach, as we target different, though related, goals:
>
> * **Difference in Core Idea:**
>     * **EchoMotion** uses the SMPL parametric prior. Our goal is to specifically improve human anatomical plausibility for complex human-centric actions.
>     * **VideoJAM** uses optical flow as its prior. Its goal is to improve general short-term temporal coherence for all pixels in any scene. It lacks a specific structural prior for the human body.
>
> * **Difference in Mechanism & Generality:**
>     * **VideoJAM** concatenates modalities at the *channel level*. This is effective but requires the modalities (video and flow) to be strictly spatially aligned in a 2D space.
>     * **EchoMotion** processes two heterogeneous modalities (pixels and parametric tokens) with different sequence lengths. They are conceptually aligned, not spatially. We concatenate them at the *sequence level* and use MVS-RoPE to align them.
>
> This "simple but effective" sequence-level approach is arguably more general, as it can handle modalities that are not spatially aligned. This is precisely what enables our unified framework to perform **cross-modal completion** (e.g., Video-to-Motion) and demonstrates its potential as a unified multi-task architecture.
>
> **2. On Quantifying the Benefits of Joint Multi-Task Training**
>
> We conducted the new quantitative experiments, specifically:
>
> * **Evaluating the Video Decoder (Motion-to-Video):** We provided a quantitative comparison of our model (in M2V mode) against specialized conditional generation methods, using Ground Truth (GT) motion as the input condition. (please see Sec. A.7.2)
> * **Evaluating the Motion Decoder (Video-to-Motion):** We also quantitatively evaluated our model's Video-to-Motion (inverse kinematics) capability, comparing its motion recovery accuracy against a baseline (e.g., the base CameraHMR estimator). (please see Sec. A.7.3)
>
> We agree this will "better highlight the advantage of joint modeling" (as the reviewer astutely noted), even if our unified model does not surpass every specialized model on every task. These new results provide a much clearer picture of the benefits of our unified approach.

---

> ### Author Response · Authors · 2025-11-21
> **Response to Review 3 Weakness (Part 2)**
>
> > **Response to Weakness 3:**
>
> We would like to clarify that LLMs were used in *two distinct stages* of our data pipeline, as detailed in Appendix A.1 of our paper:
>
> 1.  Keyword Generation: We will add the prompts used for the keyword generation to the paper's appendix (in Appendix A.4) in the final version.
>
> 3.  Video Annotation: A specialized, in-house VLM was used to generate the initial, granular textual descriptions for each video clip. To provide full transparency and to benefit the community, we will publicly release our trained VLM captioning model. To maintain the double-blind review process, we cannot provide a direct link at this time. The VLM model will be released by November 20, 2025.

---

> ### Author Response · Authors · 2025-11-21
> **Response to Review 3 Question**
>
> We sincerely appreciate the reviewer’s constructive feedback and recognition. We hope the following responses address your concerns effectively.
>
> > **Response to Question 1:**
>
> As we also discussed in our response to Weakness 1, our approach is fundamentally different from prior methods:
>
> The projection step used in prior works is where the 3D information is lost. This happens in two ways:
> - Motion Info Loss: 3D motion data is irreversibly lost during self-occlusion (e.g., a hand moving behind the back).
> - Visual Info Loss: The 2D map acts as a *strong, pixel-level control signal* that forces the video to match a simplified representation, *overriding and destroying* high-frequency visual details (like clothing, faces, and hands).
>
> Our Method (The Solution): We model the 3D SMPL parameters directly in their native parametric space via our 'Human Motion Branch'. This 3D motion data is never projected to 2D. Therefore, our model successfully overcomes the problem: it retains the full 3D motion information (even during occlusion) and preserves the high-frequency visual details of the video branch by soft constraints.
>
> > **Response to Question 2:**
>
> We thank the reviewer for this question, which allows us to clarify our tokenization process.
>
> The 51 motion tokens per frame are generated by projecting the three distinct categories of SMPL parameters, which are first grouped as described in L345. Each group is processed by an independent MLP to generate a set number of tokens.
>
> The breakdown is as follows:
>
> | Parameter Group | Original SMPL Parameters (per frame) | Resulting Tokens |
> | :--- | :--- | :--- |
> | **Shape** | $\beta \in \mathbb{R}^{10}$ (1 set of 10 shape parameters) | **1 token** |
> | **Rotation** | $\gamma \in \mathbb{R}^{6}$ (1 global 6D orientation) <br> $\theta \in \mathbb{R}^{24 \times 6}$ (24 6D joint rotations) | **25 tokens** |
> | **Position** | $v \in \mathbb{R}^{3}$ (1 3D root position) <br> $\eta \in \mathbb{R}^{24 \times 3}$ (24 3D joint positions) | **25 tokens** |
>
> In total, this process generates 1 (Shape) + 25 (Rotation) + 25 (Position) = **51 motion tokens** for each frame, which are then fed into the transformer.
>
> > **Response to Question 3:**
>
> 1. Clarification: Concatenation, not Addition
>
> First, to clarify, the motion tokens are not "added" to the visual tokens. They are concatenated along the sequence dimension to form a single, long sequence. The "diagonal arrangement" is a conceptual construct within MVS-RoPE to assign their 3D positional indices.
>
> 2. Experimental Evidence of Diagonal Arrangement:
>
> To provide direct evidence, we conducted an ablation study specifically on this "positional collision" problem, the results of which will be detailed in the Appendix. Please see Appendix Sec. A.8.1.
>
> > **Response to Question 4:**
>
> Thank you for your careful attention to detail. You are correct; this was a typo in the manuscript, and we have corrected it in the revised paper.
>
> > **Response to Question 5:**
>
> Please see our response to Weakness 2.
>
> > **Response to Question 6:**
>
> 1. Yes, we have corrected this typo in the revised paper.
> 2. Thank you for this constructive suggestion for improving the clarity of Figure 3. We agree that a clear visual distinction between "clean" and "noisy" inputs is essential. Based on your feedback, we recognize that the distinction before was not prominent enough. We have revised Figure 3, increasing the visual contrast between the clean (solid color) and noisy (textured) blocks to make the data flow in each training paradigm clearly illustrated.

---

### Official Review · Reviewer_pvjq · 2025-11-01

**Soundness:** 4
**Presentation:** 4
**Contribution:** 4
**Rating:** 8
**Confidence:** 3

**Summary:**

This paper introduces EchoMotion, a new framework designed to solve a critical problem in video generation: the synthesis of complex and kinematically plausible human motion. The authors argue that existing models, trained on pixel-only objectives, prioritize appearance fidelity at the expense of learning the underlying physical principles of human articulation, leading to anatomical artifacts and unnatural movements. To address this, EchoMotion's core idea is to model the joint distribution of video (appearance) and 3D human motion (kinematics), rather than just the video distribution conditioned on text. MVS-RoPE is proposed as a unified 3D positional encoding for both video and motion tokens and establishes an inductive bias for video-motion temporal alignment. A large-scale dataset HuMoVe with 80,000 video-motion pairs is constructed for training and achieves better human-centric video generation results.

**Strengths:**

1. The paper clearly identifies a fundamental weakness in current human-centric video generation models for kinematic correctness and proposes to explicitly model the joint distribution of video and motion as a strong inductive bias to enhance the video generation performance;
2. The MVS-RoPE design is clear and well-justified to the non-trivial problem of aligning modalities with different temporal resolutions.
3. The creation of the 80,000-pair HuMoVe dataset is a substantial contribution to the field. The lack of large-scale, high-quality, paired video and 3D motion data has been a major bottleneck.
4. The experiments are thorough and well-designed.

**Weaknesses:**

1. The paper does not provide a clear description of the specific "open-source datasets, movies, and the internet" used to build the HuMoVe dataset. Furthermore, the extracted motion could be noisy as the ground truth;
2. The framework's reliance on the SMPL model as its parametric motion representation creates an inherent bottleneck for fine-grained realism. SMPL is a whole-body model that offers very limited, or no, supervision for highly articulated and expressive areas like individual hand gestures and facial expressions.
3. Is the strong inductive bias harmful for those physical disabilities or significant bodily variations, such as amputees, as the underlying parametric model does not support this topology.

**Questions:**

I believe this paper is substantial, demonstrates improved results, and serves as a positive contribution to advancing the field of controllable video generation. Please refer to the weaknesses to further improve this paper.

---

> ### Author Response · Authors · 2025-11-21
> **Response to Review 2 Weakness**
>
> > **Response to Weakness 1: Details of Dataset Construction.**
>
> 1. Data Sourcing: The keywords were used to query large-scale, publicly available video repositories and video-sharing platforms. Our collection was restricted to human-centric clips (e.g., sports, dance, and daily activities) that were publicly accessible. We also leveraged existing open-source video datasets (e.g., those focused on human action) to increase diversity.
>
> 2. The Quality and Noise of Extracted Motion: Yes, the extracted motion could be noisy, so we use the following process to mitigate this:
> - Pre-Filtering for Visibility: Before any 3D estimation, we used DWPose to detect the number of visible keypoints. Videos with significant occlusion or where the subject was not clearly visible were filtered out, as these are the primary source of estimation failures.
> - Post-Processing: We applied temporal smoothing to the final frame-wise SMPL parameters. This step directly targets the high-frequency jitter and artifacts that are characteristic of estimation noise.
>
> > **Response to Weakness 2: Could the motion representation be a bottleneck for fine-grained realism?**
>
> We thank the reviewer for this insightful comment. This is a valid and important point: the SMPL model, by itself, does not provide supervision for expressive hand articulation or facial expressions. However, we are pleased to clarify that this limitation does not introduce a bottleneck in our framework, precisely because of our novel architectural design.
>
> - High-Level Guidance, Not Pixel-Level Control: Crucially, we do not use SMPL as a pixel-level control signal (e.g., by rendering the mesh and concatenating it at the channel level). Such an approach would indeed degrade fine-grained details, as the model would be forced to match a rendered mesh that has no face and simplified "mitt" hands.
>
> - Our Method (Sequence-Level Interaction): Instead, EchoMotion treats SMPL as a parallel stream of high-level kinematic information. This design allows the model to use SMPL to enforce the plausibility of the whole-body pose and articulation without overriding the fine-grained details.
>
> > **Response to Weakness 3: Is the inductive bias harmful for the significant bodily variations?**
>
> We thank the reviewer for this very important and insightful question.
>
> The reviewer is entirely correct. Our framework, by adopting SMPL as its motion representation, inherits the strong inductive bias of the SMPL topology. This bias is what helps the model improve anatomical plausibility for subjects that fit this specific topology.
>
> Consequently, for subjects with significant bodily variations, such as amputees, this strong bias would indeed be detrimental. The model, trained to generate outputs that align with a complete SMPL structure, would likely fail to represent such variations faithfully.
>
> We acknowledge this as a clear limitation of our current work. The primary focus of this research was to validate the effectiveness of the joint video-motion modeling paradigm for common, normative human anatomies, which is the vast majority case in most video datasets.
>
> However, we believe this is a solvable challenge and a very promising direction for future research. As the reviewer's question implies, the limitation stems from the parametric model, not the joint-modeling concept itself. We believe this could be mitigated by augmenting the motion representation with auxiliary parameters—for instance, a "topology mask" or learned embeddings that explicitly signal the absence of certain joints or limbs. This would allow the model to learn to adapt its generation, conditioning on the specific body structure.

---

### Official Review · Reviewer_ehBs · 2025-11-04

**Soundness:** 3
**Presentation:** 3
**Contribution:** 3
**Rating:** 4
**Confidence:** 3

**Summary:**

This work proposes EchoMotion that accepts both video and human motion modalities for video generation with a mixed multi-modal in-context learning strategy during the training. This work also introduces a new human-centric video dataset HuMoVe that includes paired video, 3D human motion parameters, and text data.

**Strengths:**

1. This work proposed a Dual-Modality DiT architecture that accept input and output with different modality.
2. This proposed Motion-Video Synchronized RoPE is an interesting idea to add motion information to the model.
3. This paper proposed a new high-quality dataset for video, human motion and text.

**Weaknesses:**

1. The novelty of the Dual-Modality DiT and Motion-Video Synchronized RoPE is limited. The notion of multi-modality DiT is not new and the idea of adding motion information is well studied in human mesh and skeleton generation tasks.
2. There are only baseline model results of Wan-1.3B and Wan-5B which are not enough to give accurate evaluation of the proposed architecture.
3. There is no ablation study to show the effectiveness of each proposed block in the architecture.
4. The model efficiency evaluation can add metrics like average generation fps for a more direct comparison.

**Questions:**

1. Why is there no video tuning result for Wan-1.3B at Table 1?
2. I saw there is a parameter named FPS in Table 4. Is that the FPS of the input video or something else?
3. Section 4.3 mentions that EchoMotion can operate bi-directionally. Can you provide quantitative results for this part to see the performance comparison with other state-of-the-art models?

---

> ### Author Response · Authors · 2025-11-21
> **Response to Reviewer 1 Weakness**
>
> We sincerely appreciate the reviewer’s constructive feedback and recognition. We hope the following responses address your concerns effectively. For clarity, all major revisions and newly added results in the updated PDF have been highlighted in red.
>
> > **Response to Weakness 1: Novelty of the Dual-Modality DiT and MVS-ROPE.**
>
> We respectfully clarify that while our architecture builds on established concepts, its core design, objective, and mechanisms are fundamentally different from prior art.
>
> **1. On the Dual-Modality DiT (vs. SD3's MM-DiT):**
>
> We agree that MM-DiT is an existing concept, but our application and objective are distinct.
>
> Different Core Objective: Standard MM-DiTs (e.g., SD3) model $p(x|y)$ to improve conditional generation (i.e., better prompt-following). Our Dual-Modality DiT models the **joint distribution $p(x, m|y)$**.
> * **Different Capability:** This distinct objective is what enables our model to produce novel **dual-modal outputs** (generating synchronized video *and* motion), a capability that standard MM-DiTs do not possess.
>
> **2. On Handling Heterogeneous Modalities (vs. Spatially-Aligned Data):**
>
> We address a novel challenge not present in prior multi-modal diffusion works (e.g., Wonder3D) that handle *spatially-aligned* data (like RGB + Depth maps).
>
> * **Heterogeneous Modalities:** We are the first to jointly model video pixels with **parametric motion tokens**. These modalities are *heterogeneous*—they lack direct spatial correspondence and have different sequence lengths (requiring a 4:1 temporal mapping).
> * **Purpose-Built Solution:** Our **MVS-ROPE** is the specific, novel mechanism designed to solve this non-trivial *conceptual* alignment, not just spatial. Our ablations confirm this is crucial for synchronizing the two modalities.
>
> **3. On Motion as a Parallel Modality (vs. a Control Signal):**
>
> While using motion is well-studied, prior works (e.g., RealisDance, Champ) treat it as a *unidirectional control signal* (e.g., rendering SMPL to 2D maps) to guide a video-only generator.
>
> **EchoMotion's Contribution:** EchoMotion is the first to treat parametric motion as a **co-equal, parallel modality** to be explicitly denoised alongside the video. This joint modeling of kinematics and appearance is the key to our improved anatomical plausibility and enables versatile capabilities like inverse kinematics (Video-to-Motion).
>
> > **Response to Weakness 2: Limited Baseline Comparison.**
>
> The key insight of EchoMotion is to model the joint distribution of video and parameteristic human motion, thereby enhancing the text-to-video performance on the human movement video generation. Accordingly, we compare our model mainly on its baseline video-only model to demonstrate the effectiveness of our key idea. We agree that adding more baseline comparisons could provide a more accurate evaluation of EchoMotion, so we have now introduced more open-source baseline models (please see Table 1) and will also add results from top-tier closed-source models within the next four days.
>
> > **Response to Weakness 3: More Ablation Study.**
>
> We have introduced more ablation studies in the appendix Sec A.9 to discuss the impact of positional collision in MVS-RoPE (Sec. A9.1), the effectiveness of motion-only pretraining (Sec. A9.2), and dual branch architecture design (Sec. A9.3).
>
> > **Response to Weakness 4: The FPS Matrics.**
>
> The FPS in the Table.4 refers to how many frames per second we extract from the training data for training. The selection of this parameter is consistent with the base models (Wan-1.3B, Wan-5B) to ensure a fair comparison with the base models. To further evaluate the efficiency of our model directly, we additionally present the inference time of EchoMotion compared with baseline models on a single A100 GPU here.
>
> Inference Time Comparison:
>
> | Model | Inference Time (s) |
> | :--- | :---: |
> | Wan 1.3B (Baseline) | 248.4 |
> | Wan 5B (Baseline) | 385.5 |
> | EchoMotion (on Wan 1.3B) | 285.7 |
> | EchoMotion (on Wan 5B) | 424.6 |

---

> ### Author Response · Authors · 2025-11-21
> **Response to Response to Reviewer 1 Questions**
>
> > **Response to Question 1**
>
> Thanks for your advice, we have added Wan-1.3B video tuning results in Table 1.
>
> > **Response to Question 2**
>
> Please see our response for Reviewer 1 Weakness 4.
>
> > **Response to Question 3**
>
> Thanks for your advice, we have added the qualitative results and discussions in Sec.A.7.3 and A.7.4.

---

> ### Author Response · Authors · 2025-11-25
> **Update for Reviewer 1 Weakness 2: Comparison with Top-Tier Closed-Source Models Now Included**
>
> Thank you once again for your valuable feedback.
>
> > **Update Results for Weakness 2.**
>
> As promised in our initial rebuttal, we have now completed the comparison with top-tier, closed-source commercial models in Section A.5 (Appendix). We believe these additions provide a more comprehensive evaluation of EchoMotion's capabilities and its position within the current landscape of video generation. We have uploaded the updated manuscript and sincerely hope that these new results address your concerns.

---

### Author Response · Authors · 2025-11-27
**General Reply**

Dear Committee Members and Reviewers,

We are pleased to inform you that we have replied to all the reviewers' comments. In response, we have thoroughly revised the manuscript. Key additions include the following experiments, among other improvements:

1. **Comparison with Top-Tier Models in Section A.5:** In response to [Reviewer ehBs W2, Reviewer GwaP W1, Reviewer GwaP Q4].

2. **Comparison for Motion-to-Video Task in Section A.7.2:** In response to [Reviewer ehBs Q3, Reviewer T7MT W2, Reviewer T7MT Q5, Reviewer GwaP Q2].

3. **Comparison for Video-to-Motion Task in Section A.7.3:** In response to  [Reviewer ehBs Q3, Reviewer T7MT W2, Reviewer T7MT Q5, Reviewer GwaP Q2].

4. **Evaluation of Human Motion Quality in Section A.7.4:** In response to [Reviewer GwaP W1, Reviewer GwaP Q3].

5. **Ablation Study on Motion-only Pretraining in Section A.8.2:** In response to [Reviewer ehBs W3, Reviewer GwaP Q6].

6. **Ablation Study on the Dual-Branch Architecture in Section A.8.3:** In response to [Reviewer ehBs W3].

7. **Ablation Study on the Impact of Positional Collisions in MVS-RoPE in Section A.8.1:** In response to [Reviewer ehBs W3, Reviewer T7MT Q3].

8. **Video-Tuning Ablation Results on Wan 1.3B Base Model in Table 1:** In response to [Reviewer ehBs Q1].

We would like to once again express our sincere gratitude to all reviewers for their time and constructive guidance. Your feedback has been invaluable in helping us significantly improve the quality and clarity of our work. The revised manuscript, which incorporates all the changes and additional experiments mentioned above, has been uploaded. We hope that our responses and revisions could address the initial concerns, and we look forward to further discussion.

---

### Author Response · Authors · 2025-12-02
**Rebuttal Summary for Submission 4250  (Part II)**

Our rebuttal and manuscript revisions **primarily** focused on two key areas:

1. **Experiments Enhancement:**

- **Quantitative Cross-Modal Completion Evaluation (R-ehBs, R-T7MT, R-GwaP):** We believe this is the central concern of the reviewers (R-ehBs, R-T7MT, R-GwaP). During the rebuttal, we substantially expanded our empirical validation to address this concern. We introduced new sections with quantitative metrics, discussions, and side-by-side visualizations for our model's unique Motion-to-Video (Appendix A.7.2) and Video-to-Motion (Appendix A.7.3) capabilities, validating the effectiveness of our unified framework. Experiments confirm that EchoMotion can not only perform joint generation but also serve as a versatile unified architecture, achieving competitive performance on motion-to-video and video-to-motion tasks.

- **Comprehensive Ablation Studies:** We added a full suite of ablation studies to justify our core design choices, including the Motion-Video Two-Stage Training Strategy in Section A.8.2 (R-GwaP), the spatial offset design in MVS-RoPE in Section A.8.1 (R-T7MT), and the dual-branch architecture in Section A.8.3 (R-ehBs).

- **Broader Baseline Comparisons (R-ehBs, R-GwaP):** We added extensive comparisons against both open-source models (e.g., CogVideoX family in Table 1) to show the significant improvement compared with baseline and top-tier closed-source models (Veo-3.1 and Kling 2.5 Turbo in Section A.5) to "show the gap" between them.

2. **Further Clarifications:**

 - **Core Contribution and Novelty (R-ehBs, R-T7MT):** We clarified EchoMotion's technical novelty by highlighting three key distinctions from prior work like MMDiT and VideoJAM: (1) **Objective:** It models the joint distribution of video and parametric motion, not just conditional generation. (2) **Mechanism:** It processes a unified multi-modal sequence of video and motion tokens, enabling direct multi-modal joint self-attention. (3) **Versatility:** It serves as a single, unified architecture for text-to-video-with-motion, motion-to-video, and video-to-motion tasks.
Moreover, we justified this design by discussing the limitations of alternative approaches (e.g., Cascaded Model (Motion Generator -> ControlNet) and Joint 2D Map Generation (RGB + Depth)) in response to R-T7MT Weakness-1.

 - **Training Strategy and Details (R-GwaP):** Directly responding to Reviewer GwaP's additional concerns, we have explicitly detailed the training steps, stopping criteria, and multi-task paradigm in the main paper (Section 4.1) for full reproducibility. In addition, we refined the name of the training strategy to better reflect the process: the proposed "Motion-Video Two-Stage Training Strategy," comprising Motion-only Pretraining (Phase 1) and Motion-video Multi-task Training (Phase 2).

 - **HuMoVe Dataset Construction (R-pvjq, R-T7MT):** We significantly improved the transparency of our HuMoVe dataset pipeline by: (1) detailing the initial prompts used for LLM-based keyword generation (Section A.4); (2) explaining our data sourcing and processing pipeline, including pre-filtering and post-processing methods to ensure motion quality; and (3) committing to the public release of our in-house VLM used for captioning before November 20, 2025 to ensure reproducibility.

 - **Clarification of Hyperparameters (R-ehBs):** We clarified that the "FPS" hyperparameter refers to the number of frames per second extracted from video for model training, not for inference efficiency. To further address efficiency concerns, we also provided direct inference time comparisons in our response to R-ehBs Weakness 4.

In summary, we have been highly responsive to the reviewers' feedback, culminating in a substantially improved and more rigorously validated manuscript.  We were encouraged that Reviewer GwaP (the only one to provide feedback before the halt) gave positive comments and was considering a score increase. The revised manuscript is now substantially improved and more rigorously validated. We are confident that "EchoMotion" now presents a valuable contribution to human-centric video generation, and we hope it is considered worthy of acceptance into ICLR.

Thank you for your time and dedication to the review process.

Sincerely,
The Authors of ICLR 2026 Submission 4250

---

### Author Response · Authors · 2025-12-02
**Rebuttal Summary for Submission 4250 (Part I)**

Dear Area Chair,

To facilitate your review, we provide this summary of the review and rebuttal process for our paper, "EchoMotion: Unified Human Video and Motion Generation via Dual-Modality Diffusion Transformer" (ICLR 2026 Submission 4250).

Initially, our paper received borderline scores of 8,4,4,4. While Reviewer pvjq gave an "accept" (8) with excellent ratings for contribution and soundness, other reviewers raised valid concerns, primarily regarding the need for **more comparisons on the cross-modal completion task, more ablation studies, and more details in the HuMoVe dataset construction**. In response, we conducted an extensive suite of additional experiments, clarified the potential misunderstandings, and significantly revised our manuscript to address every point.

During the discussion period, we had a productive, multi-round exchange with Reviewer GwaP, who acknowledged our progress. Crucially, the reviewer stated that "*I acknowledge that the proposed method performs well on both motion-to-video and video-to-motion tasks,*"  and "*will consider raising my score*" if the additional concerns are addressed. Unfortunately, the discussion period was halted before this could be reflected in the official score. We believe our final revisions have addressed their remaining questions.

---

### Meta-Review · Area_Chair_VqFZ · 2025-12-26

**Summary:**

Reviewers universally recognized the value of the core concept, with concerns mainly focusing on technical novelty, insufficient experimental analysis, and unclear clarification. The rebuttal effectively addressed the experimental gaps. While technical novelty is limited, I view this work as a significant pioneering exploration where the validation of the paradigm itself is the primary contribution. Although the lack of comparison with VideoJAM (or a similar baseline) is a drawback, the extensive experimental evidence provided clearly demonstrates the method's advantages. Overall, I recommend acceptance, as the paradigm and results are of significant value to the community.

**Reviewer Concerns:**

Addressed during Rebuttal:
Technical Novelty: The authors clarified that the primary innovation lies in being the first to model the joint distribution of video and parametric human motion. While some reviewers may still view the architecture as a mild innovation, its role in validating a new task paradigm is well-noted.

Generalization across Architectures: Concerns regarding base-model dependency were resolved by successfully applying the proposed method to additional models, such as CogVideoX.

Ablation Studies: The authors provided three additional groups of studies, offering a more granular understanding of their design choices.

Quantitative Validation: The dual capabilities of video-to-motion and motion-to-video were further substantiated with new quantitative evaluations in the appendix.

Qualitative Comparisons: The inclusion of side-by-side video comparisons in the supplementary materials effectively addressed the request for visual evidence against baselines.


Outstanding Points:
Comparison with Concurrent SOTA (e.g., VideoJAM): While the authors provided a conceptual discussion regarding VideoJAM, they did not include empirical results. Although VideoJAM is not open-source, a baseline re-implementation would have further strengthened the paper. However, given the strength of the other results, I do not consider this a disqualifying omission.

**Reviewer Scores:**

Reviewer ehBs (Rating: 4 $\to$ 6): The reviewer's primary concerns were addressed, leading to a positive shift in their assessment.

Reviewer pvjq (Rating: 8 $\to$ 8): Maintained a highly positive stance with no critical concerns.

Reviewer T7MT (Rating: 4 $\to$ 4): This reviewer remained hesitant to raise their score, likely due to the absence of empirical comparisons with existing conditional generation methods.

Reviewer GwaP (Rating: 4 $\to$ 6): Following multiple rounds of discussion, the reviewer acknowledged that the newly provided clarifications and updated results successfully addressed their major concerns.

---

### Decision · Program_Chairs · 2026-01-26

Accept (Poster)